# Review and syntheses: Ocean alkalinity enhancement and carbon dioxide removal through marine enhanced rock weathering using olivine

Luna J. J. Geerts[1], Astrid Hylén[1], Filip J. R. Meysman[1,2]

[1]Geobiology, Department of Biology, University of Antwerp, 2610 Wilrijk Antwerp, Belgium
[2]Department of Biotechnology, Delft University of Technology, Van der Maasweg 9, 2629 HZ Delft, The Netherlands

*Correspondence*: Luna J. J. Geerts (luna.geerts@uantwerpen.be)

**Abstract.** Marine enhanced rock weathering (mERW) is increasingly receiving attention as a marine-based carbon dioxide removal (CDR) technology. The method aims to achieve ocean alkalinity enhancement (OAE) by introducing fast-weathering rocks into coastal systems. The latter is envisioned to act as a large natural biogeochemical reactor, where ambient physical and biological processes can stimulate rock dissolution, thus generating a concomitant alkalinity release and increasing the seawater's capacity to sequester $CO_2$. Olivine has been forwarded as the prime candidate mineral for mERW, but to the present, no peer-reviewed results are available from larger scale field studies in coastal areas, so the information about olivine dissolution in marine systems is largely derived from laboratory experiments. As a result, key uncertainties remain concerning the efficiency, $CO_2$ sequestration potential, and impact of olivine-based mERW under relevant field conditions. In this review, we summarize recent research advancements to bridge the gap between existing laboratory results and the real-world environment in which mERW is intended to take place. To this end, we identify the key parameters that govern the dissolution kinetics of olivine in coastal sediments, and the associated $CO_2$ sequestration potential, which enable us to identify a number of uncertainties that are outstanding with respect to the implementation and upscaling of olivine-based ERW, as well as the monitoring, reporting, and verification (MRV). From our analysis, we conclude that the current knowledge base is not sufficient to predict the outcome of *in situ* mERW applications. Particularly, the impact of pore water saturation on the olivine dissolution rate and the question of the additionality of alkalinity generation remain critical unknowns. To more confidently assess the potential and impact of olivine-based mERW, dedicated pilot studies under field conditions are needed, which should be conducted at a sufficiently large spatial scale and monitored for a long enough time with sufficient temporal resolution. Additionally, our analysis indicates that the specific sediment type of the application site (e.g. cohesive versus permeable) will be a critical factor for olivine-based mERW applications, as it will significantly impact the dissolution rate by influencing the ambient pore water pH, saturation dynamics, and natural alkalinity generation. Therefore, future field studies should also target different coastal sediment types.

# 1 Introduction

## 1.1 Carbon dioxide removal through ocean alkalinity enhancement

Climate stabilization is a pressing challenge for society (IPCC, 2023). Scenario analysis reveals that in addition to decarbonization, active removal of carbon dioxide ($CO_2$) from the atmosphere will be required to reach the targets of the COP21 Paris agreement (IPCC, 2023; Sanderson et al., 2016; UNFCCC, 2015). Such carbon dioxide removal (CDR) has to happen fast and at a sufficiently large scale: gigaton (Gt) capacity must be reached already by 2040 and this effort should increase to 12–15 Gt $CO_2$ $yr^{-1}$ by 2100 (Rockström et al., 2017; Minx et al. 2018). As such, the need for technologies that can deliver such gigaton-scale CDR is high. However, research on the topic is currently still at an early stage, and so the efficiency, reliability, and environmental impact of most CDR techniques remain poorly constrained (Fuss et al., 2018; Minx et al., 2018; NASEM, 2022; Smith et al., 2016; Terlouw et al., 2021).

One proposed CDR technique is ocean alkalinity enhancement (OAE), which aims to increase the ocean's capacity to store $CO_2$ by raising the alkalinity level of surface waters (Hartmann et al., 2013; Renforth and Henderson, 2017). Alkalinity is the excess of proton acceptors (bases) over proton donors (acids) in solution and governs the $CO_2$ storage capacity of seawater (Dickson, 1984; Zeebe and Wolf-Gladrow, 2001). The addition of alkalinity shifts the reaction equilibria of the carbonate system from dissolved $CO_2$ towards bicarbonate ($HCO_3^-$) and carbonate ($CO_3^{2-}$), thus allowing more atmospheric $CO_2$ to dissolve in the seawater (Fig. 1) (Wolf-Gladrow et al., 2007). Alkalinity production through the weathering of silicate minerals and subsequent drawdown of atmospheric $CO_2$ is the feedback that regulates Earth's climate on geological timescales (Berner, 2004; Berner et al., 1983), and natural silicate weathering will eventually neutralize the $CO_2$ currently being released from anthropogenic activities (Archer et al., 2009). However, the timescale of this response is too slow (> 10,000 years) for society. Even if emissions were completely halted, we would have to "sit through" an extended period of global warming before the excess of anthropogenic $CO_2$ is removed naturally (Archer et al., 2009). Therefore, OAE aims to mimic the natural way by which the Earth system has responded in the geological past, but at an elevated pace. This nature-based character of OAE could help increase the societal acceptance of the CDR method (Corner and Pidgeon, 2015). Compared to other CDR approaches, OAE has the advantage that the $CO_2$ sequestration potential is considered to be substantial (>0.1–1 Gt $CO_2$ $yr^{-1}$) and that $CO_2$ storage is essentially permanent over a time scale of thousands of years (Archer et al., 2009; Caserini et al., 2021, 2022; NASEM, 2022; Renforth and Henderson, 2017). Moreover, OAE has the important benefit of counteracting ocean acidification, which is not the case for other CDR techniques that only target $CO_2$ sinking, such as reforestation on land, or blue carbon and ocean fertilization in the marine environment (Campbell et al., 2022; Caserini et al., 2022; Meysman and Montserrat, 2017).

The crux of any OAE technique relates to the source of the alkalinity, and several different OAE approaches have been suggested (NASEM, 2022). "Fast-addition OAE approaches" aim to introduce alkalinity directly to surface waters, and the alkalinity is generated either by electrochemical methods (generation of base, such as NaOH, through electrolysis of seawater) or by ocean liming (addition of nearly instantly dissolving basic minerals, such as $Ca(OH)_2$ or $Mg(OH)_2$) (Campbell et al.,

2022; Caserini et al., 2022; Eisaman et al., 2023; Rau et al., 2018; Renforth and Henderson, 2017) (Fig. 1). These technologically-oriented methods require the construction of large reactor infrastructure to produce the alkaline products that enable OAE (electrolyzer plants, lime kilns), thus necessitating substantial capital investments (NASEM, 2022; Rau, 2008;

Renforth et al., 2013). These approaches also need high amounts of energy per ton of $CO_2$ sequestered (electrochemistry) or require the installation of additional carbon capture capacity ($CO_2$ capture during lime production). In contrast, "slow-addition OAE approaches" are based on chemical mineral weathering, and are "nature-inspired" in the sense that they aim to mimic a natural process of alkalinity generation.

The idea underlying marine enhanced rock weathering (mERW) is to add specific mineral types to coastal and shelf sediments,

which then gradually dissolve over a time scale of years to centuries, thus gradually releasing alkalinity from the seabed to the overlying water (Campbell et al., 2022; Hartmann et al., 2013; NASEM, 2022). The production of rapidly weathering minerals and their addition to the seafloor requires far less energy than technology-oriented approaches, as it capitalizes on natural energy sources, such as the exergonic nature of the dissolution reaction and the *in situ* "milling" of particles using energy from waves and currents (Meysman and Montserrat, 2017; NASEM, 2022). Moreover, no large reactor infrastructure is needed, as

one essentially uses the coastal system as the biogeochemical reactor. As such, the method offers the prospect of rapid scalability, as it can be integrated within current marine engineering practices (e.g. beach nourishment, dredging, land reclamation) using existing technology and infrastructure (ports, ships, dredging equipment) (Meysman and Montserrat, 2017). A range of minerals have been considered as source material for mERW, including naturally occurring silicates (Bach et al., 2019; Hartmann et al., 2013; Lackner, 2002; NASEM, 2022; Renforth and Henderson, 2017) and carbonates (Harvey, 2008),

but also waste and overburden material (Bullock et al., 2021; Renforth, 2019; Vandeginste et al., 2024). Nevertheless, the most attention has so far been devoted to the silicate mineral olivine (Feng et al., 2017; Flipkens et al., 2023b; Fuhr et al., 2022, 2023, 2024; Griffioen, 2017; Hangx and Spiers, 2009; Hauck et al., 2016; Köhler et al., 2013; Li et al., 2024; Meysman and Montserrat, 2017; Montserrat et al., 2017; Rigopoulos et al., 2018), which is characterized by a fast intrinsic weathering rate, high $CO_2$ uptake, and large relative abundance. In this review, we synthesize the current knowledge on mERW using olivine

as a way to achieve OAE.

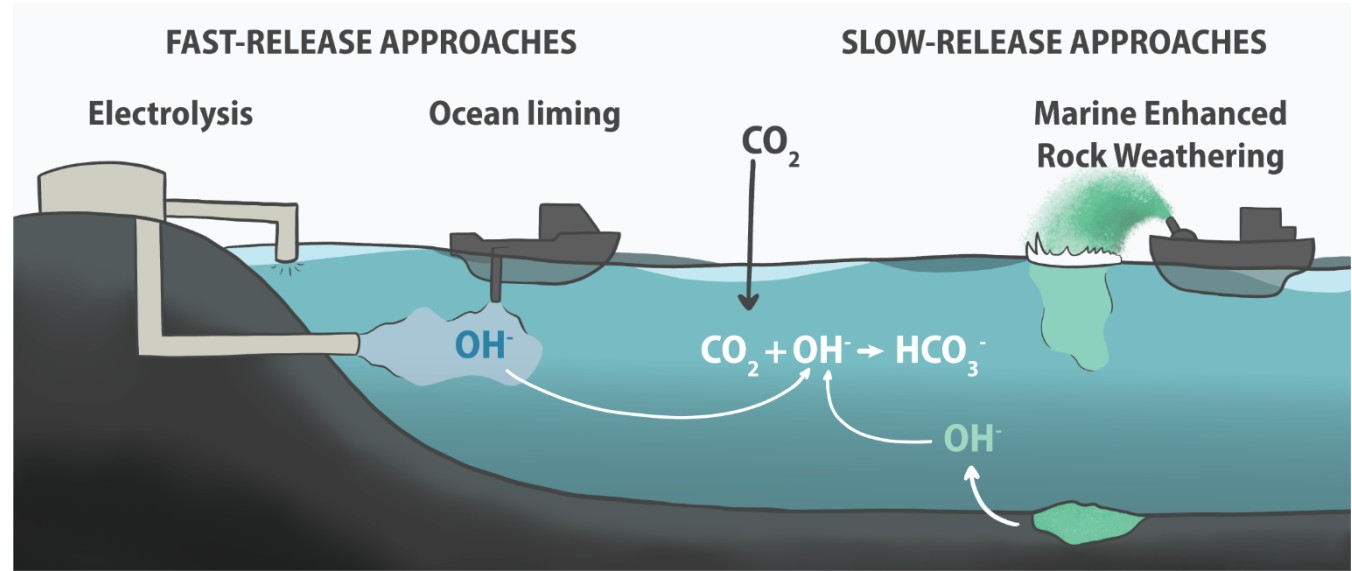

**Figure 1. Ocean alkalinity enhancement. During fast-release approaches (left), alkalinity is directly introduced into the ocean by electrolysis of seawater (which creates NaOH) or through ocean liming (addition of nearly instantly dissolving basic minerals, such as $Ca(OH)_2$ or $Mg(OH)_2$). During slow-release approaches (right), minerals (e.g., olivine) are applied to the seafloor, and alkalinity is slowly released via chemical weathering.**

### 1.2 Ocean alkalinity enhancement via enhanced rock weathering

The ultimate aim of mERW is to accelerate rock weathering, reducing the timescale of the resulting $CO_2$ uptake from millennia down to decennia. This enhanced weathering can be achieved by selectively using source rocks or minerals that have high dissolution rates, pulverizing the source rock into small particles to increase the reactive surface area, and distributing the particles in areas with high weathering potential (Meysman and Montserrat, 2017). During mERW, minerals can dissolve in the water column or be applied to the sediment. In effect, most modelling targeting the global sequestration potential have focused on rock weathering in the water column. Yet, a large energy investment is required for water column application, as particles must be ground to sufficiently small sizes (~1 µm) to stay in suspension (Köhler et al., 2013), which reduces the $CO_2$ removal efficiency significantly (up to 30%) when using fossil fuels (Hangx and Spiers, 2009; Köhler et al., 2013). Consequently, most experiments on marine mERW has targeted sediment application (Flipkens et al., 2023b; Fuhr et al., 2022, 2023, 2024; Montserrat et al., 2017; Rigopoulos et al., 2018).

A wide range of minerals has been considered for mERW. The silicate mineral olivine ($Mg_{(2-x)}Fe_xSiO_4$), particularly its Mg-rich end member forsterite, has so far received the most attention in this context due to its rapid intrinsic weathering rate (Table 1), efficient mass-to-mass $CO_2$ uptake (1.25 g $CO_2$ g fosterite$^{-1}$), and high abundance. The silicate minerals wollastonite ($CaSiO_3$; Lackner, 2002; Renforth and Henderson, 2017) and anorthite ($CaAl_2Si_2O_8$; te Pas et al., 2023) have dissolution rates that are comparable to or even higher than forsterite (Table 1). Yet, these minerals have received little attention in the context of mERW, as they are less abundant than olivine and have lower $CO_2$ capture potentials than forsterite (0.76 g $CO_2$ g wollastonite$^{-1}$ and 0.32 g $CO_2$ g anorthite$^{-1}$, te Pas et al., 2023). Carbonates (calcium carbonate: $CaCO_3$, dolomite: $CaMg(CO_3)_2$)

and brucite ($Mg(OH)_2$) dissolve faster than olivine (the nearly instantaneous dissolution of brucite could put it in the "fast-addition" OAE category; Table 1) and do not contain potentially harmful trace metals (unlike olivine, section 2.1), making them interesting minerals for mERW. However, both carbonates and brucite bear the risk of pore water oversaturation and reprecipitation, thus strongly reducing the $CO_2$ sequestration efficiency (Bach, 2024; Hartmann et al., 2013, 2022). There has been a recent interest in using brucite as a mineral for OAE via liming (e.g. Hartmann et al., 2022; Yang et al., 2023), but the cost per ton of mineral has been identified as a potential bottleneck (Kramer, 2006; Simandl et al., 2007). Rocks consisting of a multitude of minerals could also be considered for mERW. In the context of terrestrial ERW, basalt application has received considerable attention (Beerling et al., 2018), but the $CO_2$ capture potential of basalt is significantly lower than that of olivine (Rigopoulos et al., 2018). Recently, there has also been interest in using waste material, such as steel slags and mining side streams, for mERW (Bullock et al., 2021; Moras et al., 2024; Renforth, 2019). Their applicability is still under investigation, but the advantage of these materials is that they are already mined, thereby reducing the overall cost for mERW (Renforth, 2019). However, there are concerns about environmental impacts due to trace metal release from waste minerals (Bullock et al., 2021), although the first results indicate that this effect could be limited (Moras et al., 2024).

**Table 1. Dissolution times of different minerals, assuming a 1 mm radius sphere dissolving in water at pH 5 and a temperature of 25°C. Adapted from Lasaga et al. (1994) and Hartmann et al. (2013). [a]The molar volume of albite is greater than that of serpentine, giving the minerals different dissolution times for the same reactivity. The rates reported here were measured in freshwater under well-mixed conditions; in seawater and sediments, the dissolution rate may be lower due to saturation effects.**

| Mineral | Reactivity (log mol m$^{-2}$ s$^{-1}$) | Dissolution time (years) |
|---|---|---|
| Quartz[1] | -13.39 | 34,000,000 |
| Kaolinite[1] | -13.28 | 6,000,000 |
| Muscovite[1] | -13.07 | 2,600,000 |
| Epidote[1] | -12.61 | 923,000 |
| Albite[a, 1] | -12.26 | 575,000 |
| Serpentine[a, 2] | -12.26 | 533,000 |
| Sepiolite[3] | -11.85 | 79,000 |
| Enstatite[1] | -10.00 | 10,100 |
| Diopside[1] | -10.15 | 6,800 |
| **Forsterite[4]** | **-9.36** | **1680** |
| Anorthite[1] | -8.55 | 112 |
| Wollastonite [1] | -8.00 | 79 |
| Brucite[5] | -7.30 | 26 |
| Dolomite[6] | -6.70 | 1.6 |
| Calcite[6] | -5.48 | 0.1 |

[1](Lasaga et al., 1994), [2](Orlando et al., 2011), [3](Mulders et al., 2018), [4](Rimstidt et al., 2012), [5](Pokrovsky and Schott, 2004), [6](White and Brantley, 1995)

### 1.3 Marine enhanced rock weathering in different coastal environments

When mineral particles are deposited onto the sediment, they become subjected to a suite of physical, chemical, and biological processes that can stimulate their dissolution, collectively referred to as the "benthic weathering engine" (Meysman and Montserrat, 2017). This biogeochemical "engine" is primarily regulated by the hydrodynamic energy regime at a specific site, as hydrodynamics drives the transport and sorting of particles, controls the sediment transport regime, which includes physical transport processes (advection, diffusion), and key biological transport processes (biomixing, bioirrigation). While coastal

sediments represent a range of biogeochemical conditions, biogeochemical fluxes, and rates strongly correlate with the sediment type and the associated dominant transport mechanism (Aller, 2014; Silburn et al., 2017). In highly energetic systems, finer particles are transported away by erosion and deposition cycles or sorted downwards, resulting in gravel beds exposed to bedload transport. The solute transport is dominated by intense advective flushing, which gives the porewater a chemical composition similar to that of the overlying water (Aller, 2014; Huettel and Rusch, 2000; Silburn et al., 2017). When moving

to less energetic systems, smaller particles (including some organic matter) remain in place and the sediment consists mostly of sand. The sediment is permeable and advective flow is the primary solute transport mechanism, yet the smaller grain size restricts the water exchange enough to give the pore water a chemical composition that is distinct from the overlying water (Silburn et al., 2017; Widdicombe et al., 2011). In some cases, the advective water exchange can be supplemented by bioirrigation (Kristensen, 2001; Volkenborn et al., 2007). In systems with low hydrodynamic energy, fine particles settle and

form cohesive sediments rich in organic matter. The solute exchange in these sediments is driven by diffusion but is dominated by bioirrigation when larger benthic animals are present (Kristensen, 2001).

To identify the dominant controls on mERW, an abstraction into three sediment types is valuable, each with its own specific transport regime, as done in sediment transport modelling studies (e.g., Le Hir et al., 2011; Ouillon, 2018). In the *bedload scenario* (Fig. 2b), silicate sand is deposited in areas with high hydrodynamic energy and large grain sizes to promote further

physical grinding of the silicate grains during bedload transport (Meysman and Montserrat, 2017). In this scenario, minerals remain in close contact with the overlying seawater, and the weathering primarily takes place on top of the seafloor. In the *permeable sediment scenario* (Fig. 2c), silicate sand is mixed into permeable sand sediments. The advective flushing prevents the build-up of dissolution products in the sediment, which could otherwise slow down the silicate dissolution rate and cause precipitation of secondary minerals (Meysman and Montserrat, 2017). The advection of oxygen-rich water into the sediment

promotes oxic mineralization of organic matter and reoxidation of reduced compounds, which decreases the pore water pH (Rao et al., 2014; Silburn et al., 2017; Wallmann et al., 2008; Widdicombe et al., 2011), which enhances the silicate dissolution (Rimstidt et al., 2012). In the *cohesive sediment scenario* (Fig. 2d), fine silicate particles are mixed into cohesive, fine-grained sediments where biotic processes can enhance the dissolution. Bioirrigation can flush out dissolution products and introduce oxygen into the sediment which in these generally organic-matter rich sediments, leads to a large pH decrease compared to the

overlying water (Aller, 2014; Silburn et al., 2017; Widdicombe et al., 2011). Macrofauna could also speed up the dissolution of silicate minerals through ingestion due to high enzymatic activity and low pH in the guts, combined with mechanical

abrasion (Meysman and Montserrat, 2017). In cohesive sediments, the pore water pH can also be lowered by the activity of certain microbes (e.g. cable bacteria, Meysman, 2018; Pfeffer et al., 2012). As detailed below, very little *in situ* data on the strength and efficiency of the different processes contributing to this benthic weathering engine are yet available. As a result, 165 it is unclear which of the three scenarios (and hence which type of sediment locations) is the most promising for mERW.

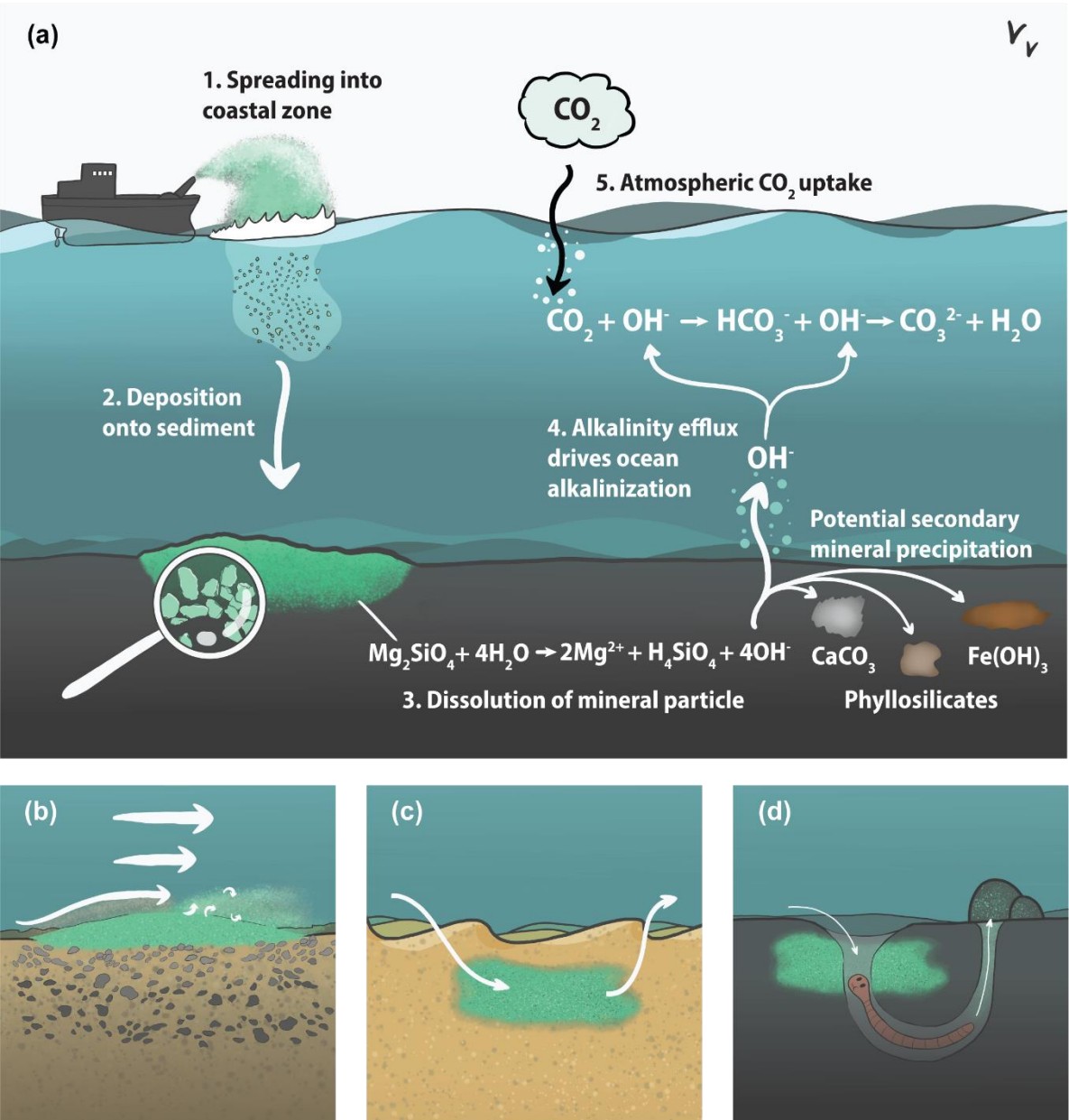

**Figure 2. Principle of marine enhanced rock weathering (mERW) as a method for ocean alkalinity enhancement, including suggested application scenarios (Meysman and Montserrat, 2017). (a) Schematic overview of mERW: (1) Finely ground particles of a fast-**

weathering silicate mineral (e.g. olivine) are spread in the coastal zone. (2) The silicate minerals are deposited onto or mixed into the sediment. (3) The silicate minerals dissolve, releasing alkalinity (here depicted as OH⁻) to the pore water. (4) Alkalinity is either transported from the sediment to the overlying water, leading to ocean alkalinity enhancement, or is trapped in the sediment by secondary reactions. (5) Upon alkalinity addition, the seawater carbonate system re-equilibrates with the atmosphere, leading to an uptake of $CO_2$. (b) Bedload application scenario. Silicate minerals are deposited on top of the sediment, allowing bedload transport to naturally grind the grains to smaller sizes. (c) Permeable sediment application. High advective flows flushing the pore water and preventing the buildup of weathering products. (d) Cohesive sediment application. Bioirrigation leads to flushing of the pore water, and oxic mineralization processes lower the pH, which increases the silicate mineral dissolution rate. The thickness of the white arrows represents the force of water movement.

## 1.4 Scope of this review

This review focuses on mERW using olivine, which is the mineral that to date has been given the most attention in this context. In recent years, the mERW literature has discussed various aspects of enhanced olivine weathering, such as rock grinding requirements (Hangx and Spiers, 2009; Strefler et al., 2018), application areas and $CO_2$ sequestration efficiency (Bertagni and Porporato, 2022; Köhler et al., 2010; Moosdorf et al., 2014), life cycle analysis (Foteinis et al., 2023), secondary precipitation and dissolution reactions (Flipkens et al., 2023b; Fuhr et al., 2022; Montserrat et al., 2017; Rigopoulos et al., 2018), olivine dissolution rates (Hangx and Spiers, 2009; Heřmanská et al., 2022; Oelkers et al., 2018; Rimstidt et al., 2012), and ecotoxicological and ecological impacts (Flipkens et al., 2021, 2023a; Guo et al., 2024; Hutchins et al., 2023; Jankowska et al., 2024; Li et al., 2024). No data is yet available from mesocosm experiments or large-scale *in situ* pilots. The first mERW mesocosm experiments and field trials are currently being planned, or are in the first stage of execution and results are being analyzed (e.g., Cornwall, 2023; USGS, 2023; Vesta, 2023),  but no peer-reviewed publications have yet emerged on their outcomes. Therefore, the information currently available on mERW is either deduced from laboratory-scale dissolution experiments or based on modeling, raising the question to what extent these results can be confidently extrapolated to a real-world application. Dissolution experiments are typically performed under high fluid-to-mineral ratios, which is the opposite of natural sediments. Several studies have investigated olivine dissolution in artificial seawater (Fuhr et al., 2022; Montserrat et al., 2017; Rigopoulos et al., 2018), while fewer studies have used natural seawater (Flipkens et al., 2023b; Montserrat et al., 2017), and only three studies include sediment (Bach, 2024; Fuhr et al., 2023, 2024). While small-scale experiments in benchtop reactors or microcosms are key to constraining specific parameters and attaining detailed process knowledge, they do not replicate the physical and biological transport and biogeochemical reactions within the natural seafloor. Therefore, a critical challenge for olivine-based mERW is the lack of information obtained from suitably large-scale experiments that are performed under *in situ* conditions, which is essential for assessing its real-world applicability (Cyronak et al., 2023; Meysman and Montserrat, 2017; Renforth and Henderson, 2017; Riebesell et al., 2023). Consequently, critical questions remain regarding the feasibility, efficiency, and ecosystem impacts of olivine-based mERW.

In this review, we summarize recent research advancements seen through the lens of practical implementation of olivine-based mERW. Compared to previous reviews on mERW (Vandeginste et al., 2024; Wang et al., 2023), we concentrate on the context of coastal sediments since the conditions within this application environment are central to the outcome of mERW. Our focus is on predictability: given a target site in the coastal ocean, how confidently can we predict the olivine dissolution rate and the

205 associated rate of alkalinity release and $CO_2$ sequestration? What aspects and parameters need consideration, and which are currently uncertain (Table 2)? What aspects need specific consideration when performing future mesocosm pilots and field trials? Compared to previous work on mERW, we specifically consider the heterogeneity of sediments within the coastal ocean and hence the impact of sediment texture. We direct our attention to the various parameters that determine mineral dissolution rates in different types of coastal sediments and we evaluate the ensuing $CO_2$ sequestration during olivine-based mERW. Our

goal is to bridge the gap between the experimental laboratory studies that have been conducted so far and the real-world environment. We review the importance of processes that could hamper the efficiency of mERW (e.g., cation-depleted surface layers, secondary mineral formation, pore water saturation) and highlight aspects that demand further scrutiny in future experiments and field trials. Finally, we discuss some issues relevant to efficient monitoring, reporting, and verification (MRV) of mERW applications.

**Table 2. Overview of parameters and terminology used in the manuscript.**

| Symbol | Parameter | Value |
| --- | --- | --- |
| $A_0$ | Pre-exponential factor Arrhenius equation (mol m$^{-2}$ s$^{-1}$) | |
| $A_{BET}$ | BET measured specific surface area (m$^2$ g$^{-1}$) | |
| $A_{geo}$ | Geometric specific surface area (m$^2$ g$^{-1}$) | |
| $A_{surf}$ | Specific surface area (m$^2$ g$^{-1}$) | 2–8 |
| $a_{H_2O}$ | Water activity (unitless) | |
| $C_{mineral}$ | Mineral content of sediment at a given time (g mineral m$^{-2}$ seafloor) | |
| $D_e$ | Equivalent grain diameter (m) | |
| $D_{max}$ | Maximum grain size in grain size interval (m) | |
| $D_{min}$ | Minimum grain size in grain size interval (m) | |
| $E_a$ | Activation energy of olivine dissolution in Arrhenius equation (kJ mol$^{-1}$) | 70.4 (pH < 5.6), 60.9 (pH > 5.6) [1] |
| $k_d$ | Intrinsic dissolution rate constant of olivine (mol m$^{-2}$ s$^{-1}$) | |
| $L_{mineral}$ | Mineral loading, $C_{mineral}$ at $t_0$ (g mineral m$^{-2}$ seafloor) | |
| $M_{CO_2}$ | Molar mass of $CO_2$ (g mol$^{-1}$) | 44.01 |
| $M_{FAY}$ | Molar mass of fayalite (g mol$^{-1}$) | 203.77 [2,3] |
| $M_{FOR}$ | Molar mass of forsterite (g mol$^{-1}$) | 140.69 [2,3] |
| $M_{olivine}$ | Molar mass of olivine (g mol$^{-1}$) | 140.69–203.77 [2,3] |
| $P_{CO_2}$ | $CO_2$ capture potential (g $CO_2$ sequestered g$^{-1}$ dissolved olivine) | |
| $R$ | Universal gas constant (kJ K$^{-1}$ mol$^{-1}$) | 8.314 x 10$^{-3}$ |
| $F_{mineral}$ | Areal mineral weathering rate (g mineral dissolved m$^{-2}$ seafloor yr$^{-1}$) | |
| $F_{A_T}$ | Areal alkalinity release rate (mol alkalinity m$^{-2}$ seafloor yr$^{-1}$) | |

| $F_{CO2}$ | Areal $CO_2$ sequestration rate (g $CO_2$ m$^{-2}$ seafloor yr$^{-1}$) | |
|---|---|---|
| $R_{diss}$ | Specific mineral dissolution rate (g mineral dissolved g$^{-1}$ mineral present yr$^{-1}$) | |
| $R_S$ | Grain roughness (unitless) | 2–8 |
| $T$ | Temperature (K, C°) | |
| $V_{olivine}$ | Molar volume of olivine (m$^3$ mol$^{-1}$) | $4.365 \times 10^{-5}$[2] |
| $x_{FAY}$ | Mole fraction fayalite in olivine (unitless) | 0.07–0.20 |
| $x_{inert}$ | Mass fraction inert minerals in dunite source rock (unitless) | 0–0.1 |
| $\eta_{A_T}$ | Alkalinity transfer efficiency (unitless) | 0–1 |
| $\gamma_{A_T}$ | Alkalinity production factor (unitless) | 0–1 |
| $\rho_{CO_2}$ | $CO_2$ sequestration efficiency (mol $CO_2$ mol$^{-1}$ alkalinity) | 0.75–0.90[4,5] |
| $\varphi$ | Fraction forsterite in particular grain diameter class | |
| $\Omega$ | Saturation index of mineral dissolution reaction (unitless) | 0–1 |

[1](Rimstidt et al., 2012), [2](Deer et al., 2013), [3](Flipkens et al., 2021), [4](Bertagni and Porporato, 2022), [5](Schulz et al., 2023)

## 2 Olivine rocks and their availability

### 2.1 Olivine composition and source rocks

Olivine ($Mg_{(2-x)}Fe_xSiO_4$) constitutes a common group of igneous minerals, with compositions ranging from the magnesium endmember forsterite ($Mg_2SiO_4$, $M_{FOR}$ = 140.69 g mol$^{-1}$) to the iron endmember fayalite ($Fe_2SiO_4$, $M_{FAY}$ = 203.77 g mol$^{-1}$) (Deer et al., 2013; Kremer et al., 2019). The molar mass of olivine ($M_{olivine}$) hence reflects the ratio between forsterite and fayalite, with $M_{oli} = (1 - x_{FAY})M_{FOR} + x_{FAY}M_{FAY}$, where $x$ is the molar fraction of fayalite. Olivine-rich rock (dunite) also contains trace metals, most notably, nickel (Ni) and chromium (Cr). Ni substitutes the divalent cations in olivine ($Mg^{2+}$, $Fe^{2+}$),

and its content ranges from 0.2–1.2 mol% (Keefner et al., 2011; Montserrat et al., 2017; Santos et al., 2015). Cr is typically present as chromite ($FeCr_2O_4$) inclusions at lower concentrations of 0.02–0.66 mol% (Deer et al., 2013; Flipkens et al., 2021). The fate of these metals upon dissolution and their potential impact on marine ecosystems remains an important topic of research for mERW (Flipkens et al., 2021, 2023a; Foteinis et al., 2023; Guo et al., 2024; Hutchins et al., 2023; Jankowska et al., 2024), but falls outside of the scope of this review.

Olivine is one of the most rapidly weathering silicate minerals (Table 1) due to its structure, which consists of independent silicate tetrahedra ($SiO_4$) linked by relatively weak Mg/Fe-O bonds (Sun and Huggins, 1947). This structure differs from most other silicate minerals, in which $SiO_4$ tetrahedra are connected through a Si-O-Si bond, which has a three times greater bond strength and hence is much harder to break (Velbel, 1999). Once the metal ion ($Mg^{2+}/Fe^{2+}$) in olivine is mobilized, the $SiO_4$ tetrahedra are also liberated and move into solution as dissolved orthosilicic acid, $Si(OH)_4$ (Oelkers et al., 2018).

Olivine is a major constituent of many ultramafic and mafic igneous rocks such as gabbro, peridotite, and basalt, where it coexists with plagioclase and pyroxene (Deer et al., 2013; Klein et al., 2002). Because olivine is so easily altered by weathering,

it is not commonly found at the Earth's surface (Delvigne et al., 1979; Wilson, 2004). The highest concentration of olivine is found in peridotite, which is an umbrella term for ultramafic rocks containing at least 40 weight% olivine (Fig. 3a). Peridotite is further divided into dunite, harzburgite, wehrlite, and lherzolite, based on the relative abundance of olivine relative to orthopyroxene and clinopyroxene (Fig. 3a). Dunite contains over 90 weight% olivine by definition, and is, therefore, the most relevant peridotite rock for mERW (Caserini et al., 2022; Deer et al., 2013; Le Maitre et al., 2002). When using dunite as the olivine source rock, the mass fraction of inert minerals ( $x_{inert}$ ) thus ranges between 0 and 0.10.

Dunite rocks with a high forsterite content are preferred for mERW. The ferrous iron ($Fe^{2+}$) produced when fayalite dissolves, will precipitate back as iron ($Fe^{3+}$) (hydr)oxides upon contact with $O_2$, which consumes any alkalinity produced during fayalite dissolution (Griffioen, (2017), section 3.2.2). Therefore, fayalite does not contribute to $CO_2$ drawdown. Typically, the forsterite fraction of olivine is substantially higher than the fayalite fraction, with $x_{FOR}$ values of 0.80–0.88 in Fe-rich dunites and 0.88–0.94 in Mg-rich dunites, respectively (Ackerman et al., 2009; Deer et al., 2013; Harben and Smith, 2006; Rehfeldt et al., 2007; Su et al., 2016).

## 2.2 Availability of olivine

Olivine deposits are typically found at ultramafic intrusions (e.g. Norway, Germany), ophiolite complexes (e.g. Greece, Italy, Turkey), alpine peridotites emplaced along thrust faults (e.g. Italy, Spain), rift zones/ basalts of mid-ocean ridges (e.g. Iceland), and volcanic xenoliths (e.g.German Eifel and Kaiserstuhl, Iceland) (Harben and Smith, 2006; Kremer et al., 2019) (Fig. 3b–3c). Large dunite reserves are found within the Fjordane Complex, Norway (>2 Gt), the Piedmont region in Italy (650 Mt), Horoman Hill in Japan (100 Mt), and the Xixia and Yubian Counties in China (9 Mt) (Caserini et al., 2022; Harben and Smith, 2006). In the United States, a large dunite deposit (200 Gt) is present at Twin Sister Mountain (Caserini et al., 2022; Goff et al., 2000; Kremer et al., 2019), and a smaller one (300 Mt) in North Carolina (Caserini et al., 2022; Goff and Lackner, 1998). Dunite deposits are commercially exploited at several locations across the globe, as olivine is used as a slag conditioner in steel production to improve the performance and lifespan of the steel melting furnace (Harben and Smith, 2006). Established mining reserves for dunite amount to a few tens of gigatonnes (Gt), while potential resources are estimated at a few hundreds of Gt (see supplementary file 2, Harben and Smith 2006, and Caserini et al. 2022). For reference, the global reserves for wollastonite have been estimated at ~500 Mt, with purities of 40–96 weight% (Robinson et al., 2006), and are hence considerably smaller than those of olivine. At a theoretical $CO_2$ capture potential of 1.25 g $CO_2$ per g of forsterite dissolved (Hartmann et al., 2013), the currently exploited dunite reserves translate to a total CDR capacity of >50 Gt $CO_2$, which can increase up to a few hundred Gt if new mining deposits are exploited. The CDR potential for olivine could thus be considerable but note that the total CDR requirement over the 21st century to reach the Paris agreement targets is estimated at 400–1000 Gt $CO_2$ (Minx et al., 2018; Riahi et al., 2021; Rockström et al., 2017), which underlines to the extraordinary scale of the climate challenge. Therefore, olivine-based mERW should be considered within a portfolio of parallel CDR approaches. The global production rate of olivine sand is currently ~8 Mt per year (Harben and Smith, 2006), so this production rate would have to increase substantially to enable large-scale mERW. Ideally, olivine for mERW should be sourced close to the deployment

location, as large transport distances considerably impact the environmental sustainability of mERW (Foteinis et al., 2023). The type of transport also matters, with overland transport via road and rail being 9 and 3 times more $CO_2$ intensive than maritime transport (Renforth, 2012). Potential prime candidate sources for olivine rock are therefore large mines close to the sea (e.g. Åheim mine – Norway).

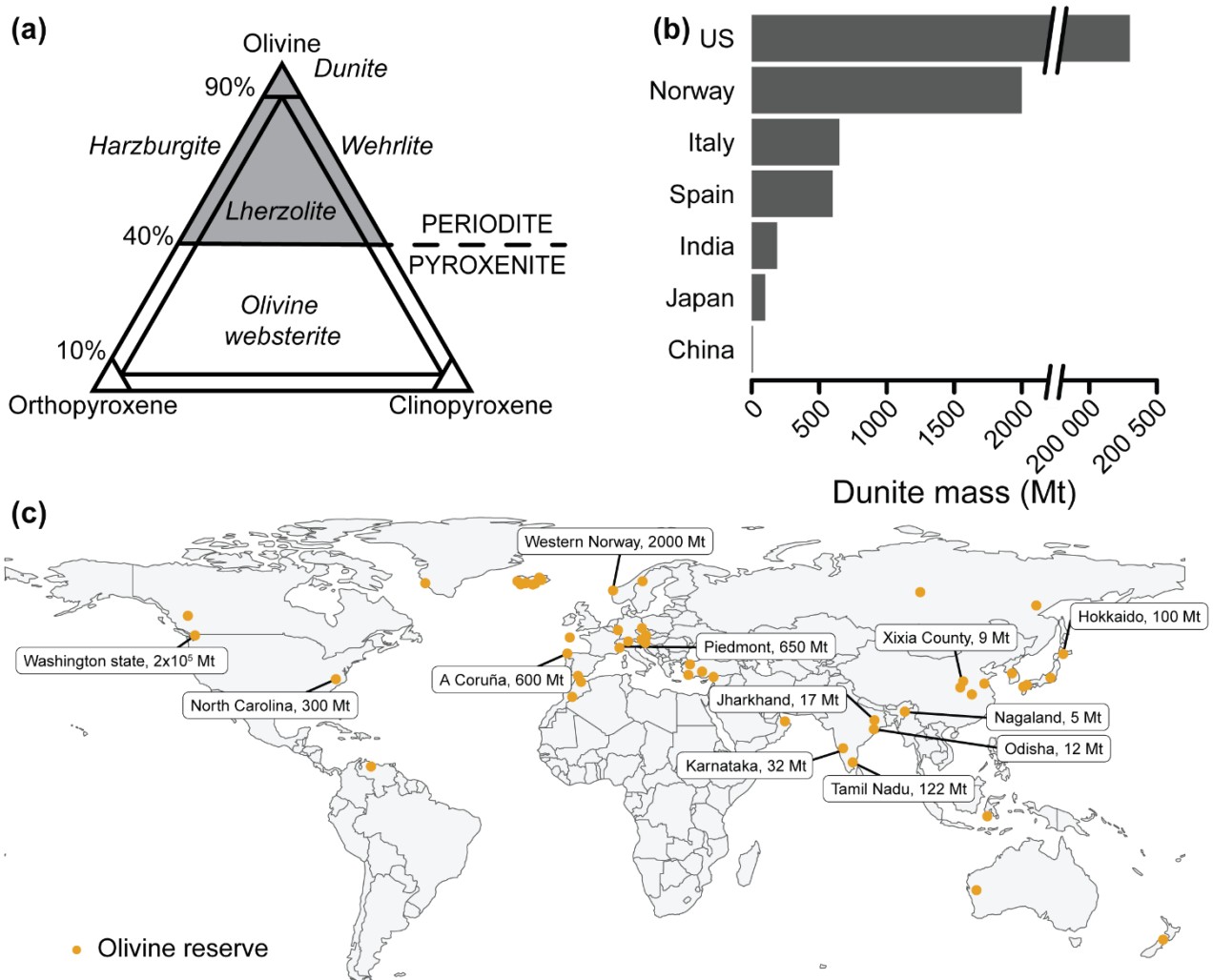

**Figure 3. Olivine classification and occurrence around the globe. (a) Ternary plot for rock composed of olivine, orthopyroxene, and clinopyroxene. The peridotite group (grey) contains at least 40 weight% olivine. Dunite contains at least 90 weight% olivine. (b) The seven countries with the largest known dunite reserves. The best available reservoir estimate is given; the precise amount is uncertain. (c) Global map of locations with known dunite deposits (non-exhaustive). When known, the size of the deposit is displayed. A list of references and details is found in supplementary file 2.**

# 3 Factors affecting the CO₂ sequestration rate during mERW

The entire process of mERW can be quantitatively described as a six-step process (Fig. 2a): (1) production of olivine sand from olivine rich source rock (2) transport of the project site, (3) spreading and deposition onto the seabed (4) mineral dissolution, (5) alkalinity release to the overlying water, and (6) $CO_2$ sequestration at the air-sea interface. In the dissolution step, the olivine sand deposited during mERW will react on top of or within the seabed. The areal weathering rate (g mineral dissolved m⁻² seafloor per unit of time) is provided by the rate expression (Meysman and Montserrat, 2017):

$$F_{mineral}(t) = -R_{diss}(t)C_{mineral}(t) \qquad (1)$$

The specific mineral dissolution rate $R_{diss}$ denotes the amount of source mineral that is lost per unit of time (g mineral dissolved g⁻¹ mineral present yr⁻¹), while the mineral content $C_{mineral}$ represents the amount of olivine sand that is present per unit of application area at a given time $t$ (g mineral m⁻² seafloor). The mineral loading $L_{mineral} = C_{mineral}(t_0)$ represents the amount of olivine sand initially deposited. When feedstock is utilized that contains an assemblage of different alkalinity-producing minerals (e.g. basalt), each mineral will be defined by its specific dissolution rate $R_{diss}$. The overall $R_{diss}$ for the feedstock can be calculated by weighted averaging of the mineral-specific $R_{diss}$ values using the mass ratios.

The weathering process will generate a certain amount of alkalinity ($A_T$) that escapes across the sediment-water interface towards the overlying seawater. The areal alkalinity release rate (mol alkalinity m⁻² seafloor per unit of time) is linked to the areal weathering rate by:

$$F_{A_T} = 4\gamma_{A_T}\left(-\frac{1}{M_{olivine}} \cdot (1-x_{inert}) \cdot F_{mineral}\right) \qquad (2)$$

The alkalinity production factor $\gamma_{A_T}$ (unitless) denotes how much alkalinity is produced upon mineral dissolution, relative to complete dissolution of olivine (producing 4 mol alkalinity per mol olivine). This factor depends on the mechanism of dissolution (different end products of olivine weathering are possible) and any loss of alkalinity due to precipitation of secondary minerals (section 3.2). The factor $x_{inert}$ represents the mass fraction of inert minerals in the source rock, i.e., accessory minerals that do not dissolve and generate alkalinity, while $M_{olivine}$ denotes the molar mass of olivine. The areal alkalinity release rate $F_{A_T}$ thus represents the alkalinity that effectively reaches the overlying water.

In the final step, $CO_2$ sequestration will occur at the air-sea interface. The areal $CO_2$ sequestration rate (g $CO_2$ m⁻² seafloor per unit of time) can be described as:

$$F_{CO_2} = M_{CO_2} \cdot \rho_{CO_2} \cdot \eta_{A_T} \cdot F_{A_T} \qquad (3)$$

Here, $M_{CO_2}$ represents the molar mass of $CO_2$ (44.01 g mol⁻¹), $\rho_{CO_2}$ denotes thermodynamic $CO_2$ sequestration efficiency, i.e., the amount of $CO_2$ sequestered upon adding one mole of alkalinity to seawater (mol $CO_2$ mol⁻¹ alkalinity), and $\eta_{A_T}$ represents the alkalinity transfer efficiency, the fraction of alkalinity that effectively equilibrates with the atmosphere (coastal waters can be downwelled to the deep ocean before full $CO_2$ equilibration with the atmosphere has taken place). By combining Eq. 1–3, the $CO_2$ uptake rate can be expressed as:

$$F_{CO_2} = P_{CO_2}(t)R_{diss}(t)C_{mineral}(t)$$

The $CO_2$ capture potential $P_{CO_2}$ (g $CO_2$ sequestered g$^{-1}$ dissolved mineral) specifies the mass of $CO_2$ sequestered from the atmosphere per unit mass of source rock dissolved and is formally defined as (Meysman and Montserrat, 2017):

$$P_{CO_2} = \frac{M_{CO_2}}{M_{olivine}} \cdot \rho_{CO_2} \cdot \eta_{A_T} \cdot 4\gamma_{A_T} \cdot (1 - x_{inert}) \tag{4}$$

In the following sections, we will systematically review all the parameters that control the $CO_2$ capture potential $P_{CO_2}$ and mineral dissolution rate $R_{diss}$. All parameters are expressed for olivine-based mERW (summarized in Table 2).

### 3.1 Olivine dissolution rate

The olivine dissolution rate $R_{diss}(t)$ can be parameterized as:

$$R_{diss}(t) = k_d \cdot A_{surf}(t) \cdot M_{olivine} \cdot (1 - \Omega(t)) \tag{5}$$

Here, $k_d$ is the intrinsic dissolution rate constant (mol olivine dissolved m$^{-2}$ of olivine surface area yr$^{-1}$), $A_{surf}$ is the specific surface area at a given time (m$^2$ g$^{-1}$), $M_{olivine}$ is the molar mass of olivine (g$^{-1}$ mol), and $\Omega$ is the saturation state. Saturation occurs when dissolution products build up in the pore water, eventually reaching thermodynamic equilibrium ($\Omega = 1$ meaning full saturation) and thus impeding further dissolution (Köhler et al., 2010). Montserrat et al. (2017) and Flipkens et al. (2023b) reported that saturation effects slowed down dissolution rates in some of their long-term laboratory experiments. Yet, little is
known about how the saturation state impacts olivine weathering in the pore water of marine sediments. The three mERW application scenarios considered (Fig. 2b–d) all assume some form of continuous exchange of the (pore) water surrounding the olivine, thereby preventing saturation effects. However, saturation effects are expected to occur in cohesive sediments with little advection or biological irrigation, or when dissolution rates are very high (e.g. when small grain sizes are used). Potential saturation effects on olivine dissolution in marine sediments hence comprise an important aspect of investigation in future
studies. In the following sections, we discuss the factors affecting $k_d$ and $A_{surf}$, and their effects on the olivine dissolution rate.

### 3.1.1 The intrinsic dissolution rate of olivine

Many studies have investigated the $k_d$ value of forsterite in aqueous solutions with a high fluid-to-mineral ratio, thus enabling experimental conditions to remain far from thermodynamic equilibrium (Bailey, 1976; Blum and Lasaga, 1988; Chen and
Brantley, 2000; Eriksson, 1982; Giammar et al., 2005; Golubev et al., 2005; Grandstaff, 1986; Hänchen et al., 2006, 2007; Hausrath and Brantley, 2010; Luce et al., 1972; Oelkers, 2001a; Olsen et al., 2015; Olsen and Donald Rimstidt, 2008; Pokrovsky and Schott, 2000; Prigiobbe et al., 2009; Rosso and Rimstidt, 2000; Shirokova et al., 2012; Siegel and Pfannkuch, 1984; Van Herk et al., 1989; Wogelius and Walther, 1991). The resulting data have been synthesized in several reviews (Heřmanská et al., 2022; Oelkers et al., 2018; Rimstidt et al., 2012). This interest in olivine dissolution kinetics is due to its
relatively simple reaction mechanism, its potential role in $CO_2$ sequestration (Heřmanská et al., 2022; Oelkers et al., 2018;

Rimstidt et al., 2012), and more recently in reconstructing the past climate on Mars (Gaudin et al., 2018; Hausrath and Brantley, 2010; Niles et al., 2017; Olsen et al., 2015). These studies show that $k_d$ primarily is affected by pH and temperature (Brantley et al., 2008; Crundwell, 2014; Heřmanská et al., 2022; Olsen, 2007; Pokrovsky and Schott, 2000; Rimstidt et al., 2012; Wogelius and Walther, 1991). The following empirical equations for the dissolution rate constant, valid between $0 < pH < 14$ 

and $0°C < T < 150°C$, have been provided (Heřmanská et al., 2022; Rimstidt et al., 2012):

$$\log(k_d) = 5.17(0.16) - 0.44(0.01)pH - 3675(47.0)1/T \qquad (pH < 5.6) \qquad (6)$$

$$\log(k_d) = 2.34(0.40) - 0.22(0.02)pH - 3179(143)1/T \qquad (pH > 5.6) \qquad (7)$$

It is important to note that $k_d$ values are normalized to the specific surface area of the olivine grain (section 3.1.5), which is represented in two ways: either the actual surface area as estimated via Brunauer, Edward, and Teller (BET) analysis ($A_{BET}$)

or the geometrical surface area ($A_{geo}$), which assumes that the dissolving grain is a perfect sphere. The choice of surface normalization procedure critically influences the resulting $k_d$ values, as $A_{BET}$ can be considerably larger than $A_{geo}$. The Rimstidt relations Eq. (6) and Eq. (7) are normalized using $A_{BET}$, so the resulting $k_d$ values should not be compared with those normalized using $A_{geo}$ without proper conversion.

Here, we discuss the impact of pH and temperature on $k_d$ (Fig. 4a), as well as the effect of salinity, due to its relevance for

coastal environments. Some studies have also suggested that the $CO_2$ concentration can affect the dissolution rate of olivine at a pH higher than 6 (Pokrovsky and Schott, 2000; Wogelius and Walther, 1991). However, more recent studies found no explicit $CO_2$ effect when correcting for the change in pH caused by elevated $CO_2$ concentrations (Golubev et al., 2005; Prigiobbe et al., 2009). As such, $CO_2$ appears to affect olivine dissolution only indirectly through a change in pH. Rates of $k_d$ in Fig. 4 were compiled from Rimstidt et al. (2012) and Heřmanská et al. (2022), supplemented with additional data (Flipkens et al.,

2023b; Gerrits et al., 2020; Hausrath and Brantley, 2010; Lunstrum et al., 2023; Montserrat et al., 2017; supplementary file 2). The 95% confidence interval (CI) around the value of $k_d$ is large, spanning an order of magnitude at pH < 5.6 and several orders of magnitude at pH > 5.6 (Fig. 4a). This large spread in the data has previously been discussed by Oelkers et al. (2018) and Rimstidt et al. (2012), who attributed it to mineral purity (pure olivine dissolves faster e.g. Golubev et al., 2005), initial incongruent dissolution of olivine, lack of common data format and inconsistent reporting, differences in sample grinding and

preparation, and most importantly, to inaccuracies in reactive surface area measurements (section 3.1.5). The problem of incongruent dissolution as well as the formation of secondary clay minerals could be potentially constrained through silicon isotope analysis (Chemtob et al., 2015; Gruber et al., 2013), if the source dunite rock has an isotope that is sufficiently distinct

from the silicate sources in the application site sediment. However, such isotope analysis has not yet been performed in mERW studies and could be an avenue for future research.

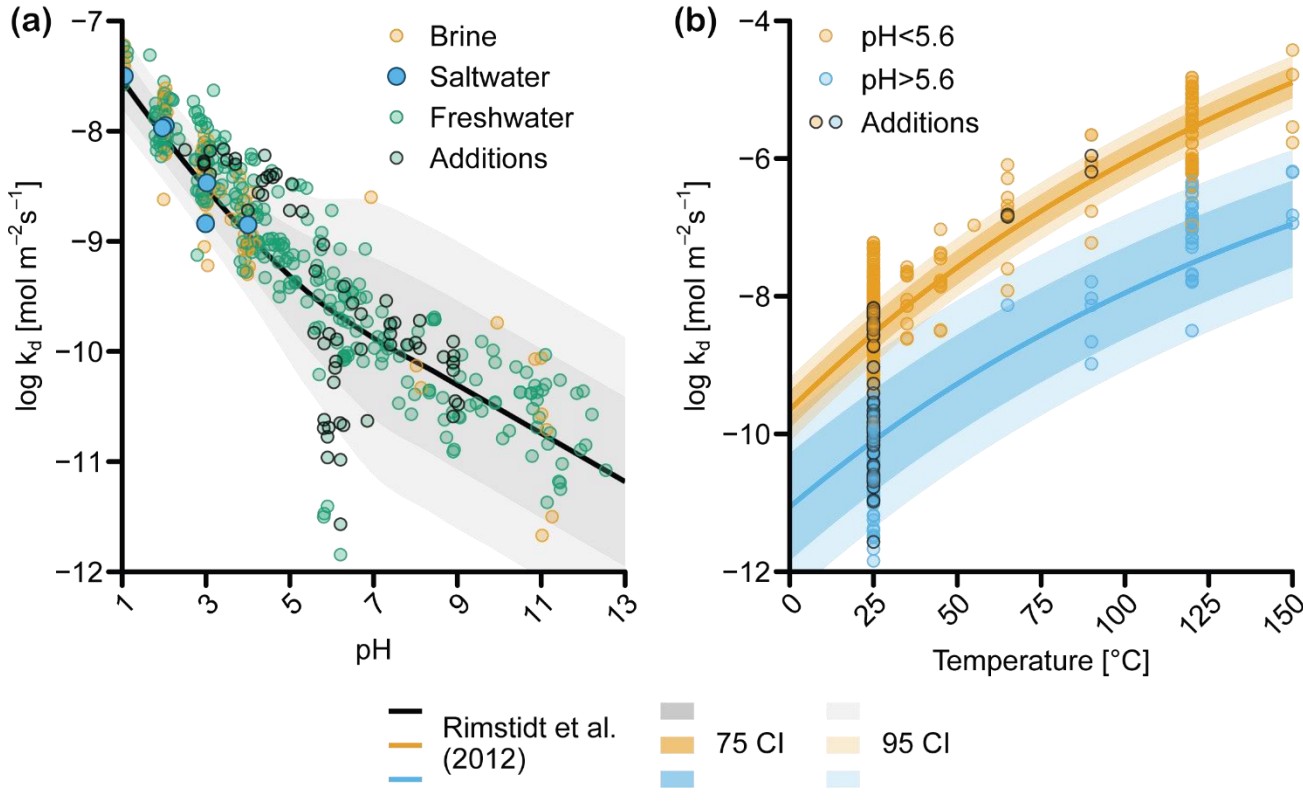


**Figure 4. (a) Forsterite dissolution rates ( $k_d$ ), adjusted to 25°C, as a function of pH according to the rate equations from Rimstidt et al. (2012). Data are normalized to BET surface areas and categorized according to salinity (freshwater < 0.6 mol l$^{-1}$, saline = 0.6– 0.85 mol l$^{-1}$, brine > 0.85 mol l$^{-1}$). Shadings show confidence intervals (CI) derived using the standard error for each parameter (supplementary file 1.2). (b) $k_d$ as a function of temperature. The rate equation was solved for pH 3.13 and 8.22, respectively, which** 375 **were the average pH values of experiments that Rimstidt et al. (2012) classified as "acidic" (pH < 5.6) and "basic" (pH > 5.6). "Additions" marks data from dissolution experiments where factors other than pH and temperature were investigated (e.g., the addition of dissolved organic matter, the addition of aluminum), which may have affected $k_d$ (supplementary file 1.1).**

### 3.1.2 Impact of pH on olivine dissolution

The olivine dissolution rate constant $k_d$ follows a log-linear dependency on pH (Fig. 4a). Different mechanisms have been
suggested to explain the observed change in dissolution rate between acidic and basic conditions. Pokrovsky and Schott (2000) argued that different reaction mechanisms occur at pH < 9 and pH > 9. For acidic and slightly alkaline conditions (pH < 9), Mg$^{2+}$/Fe$^{2+}$ ions within the olivine are exchanged with two protons, creating a silica-rich, Mg/Fe-free protonated surface precursor complex followed by the sorption of one proton on two polymerized silica tetrahedra (Pokrovsky and Schott, 2000). A lower pH increases the concentration of free protons in solution, which can exchange with Mg/Fe in the olivine and accelerate

the release of silica tetrahedra and hence the dissolution of olivine (Oelkers et al., 2018; Pokrovsky and Schott, 2000). In basic conditions (pH > 9), silica would be preferentially released, and Mg/Fe sites hydrated, forming $(Mg/Fe)OH_2^+$ species that control the dissolution (Pokrovsky and Schott, 2000).

Rimstidt et al. (2012) also proposed two separate pH regimes, but with a transition at pH 5.6 rather than pH 9 (Fig. 4a). The dependency of $k_d$ on pH is twice as strong at pH < 5.6 compared to pH > 5.6. While Rimstidt et al. (2012) point out that the

apparent break in the data could be due to the relative paucity of rate measurements for pH > 6 and temperatures below 25°C, they also suggest that different mechanisms may drive the dissolution at low and high pH, leading to the two rate reactions presented in Eq. (6) and Eq. (7) (Rimstidt et al., 2012). Based on the dataset from Rimstidt et al. (2012), Crundwell (2014) proposed that an additional step occurs in basic and slightly acidic conditions (pH > 5.6), whereby $H^+$ ions absorb at the inner Helmholtz layer of the silicate mineral. This absorption would enable the release of Mg/Fe to the solution, after which the

adsorbed $H^+$ ion can react with the silica tetrahedra to form $H_4SiO_4$, which goes into solution. This reaction mechanism explains the slope change at a pH~5.6 (Fig. 4a). In conclusion, there seems to be experimental and theoretical support for two dissolution mechanisms of olivine depending on pH regime, with considerably higher dissolution rates at lower pH (< 5.6).

The range of pH conditions relevant to mERW under field conditions is relatively narrow compared to the range shown in Fig. 4a. While the pH of coastal seawater is relatively high and shows minor variation (pH ≈ 7.9–8.3), the pH in coastal sediments

is generally more acidic and also variable with depth (pH ≈ 6–8) (Cai et al., 1995; Pfeifer et al., 2002; Rao et al., 2012; Silburn et al., 2017; Widdicombe et al., 2011; Zhu et al., 2006). Olivine is therefore expected to show higher dissolution rates in the sediment compared to the water column (Meysman and Montserrat 2017). However, accurate descriptions of the spatial and temporal variability of the pore-water pH are currently lacking (Silburn et al., 2017). The pore-water pH is determined by microbial redox reactions, as well as mineral dissolution and precipitation, which in turn are controlled by sediment

characteristics and the presence of burrowing fauna. Certain microbes, such as cable bacteria, can drastically change the pH profile (Meysman, 2018; Pfeffer et al., 2012) and have been found in previous mERW studies where they were suggested to lower the pH and increase the dissolution of $CaCO_3$ (Fuhr et al., 2023). Within the same sediment, spatial and temporal pH variations are the result of seasonal changes in the organic matter supply, which affects the sedimentary $O_2$ demand (Silburn et al., 2017; Widdicombe et al., 2011), alters the physical advection and biological irrigation (Meysman et al., 2007), and

changes the microbial activity (Meysman et al., 2015). In addition to being a master variable for olivine dissolution, pH also determines the possibility of secondary reactions (e.g. sepiolite formation). The lack of data on olivine dissolution rates at pore-water pH in marine conditions and the absence of suitable spatial maps of sediment pH within coastal environments emerge as important research gaps that hamper the prediction of $CO_2$ sequestration by mERW. Careful monitoring of sediment pH during future field applications is required to designate the best application areas for mERW.

**3.1.3 Impact of temperature on olivine dissolution**

The $k_d$ value increases with temperature, and this relationship is classically described via the Arrhenius equation (Casey and Sposito, 1992; Oelkers et al., 2018):

$$k_d = A_0 \exp\left(\frac{-E_a}{RT}\right) \qquad (8)$$

where $A_0$ refers to the temperature-independent pre-exponential factor (mol m$^{-2}$ s$^{-1}$), $E_a$ stands for the apparent activation energy of olivine dissolution (J mol$^{-1}$), $R$ is the gas constant (J K$^{-1}$ mol$^{-1}$), and $T$ is the temperature (K). The Arrhenius equation can be used to rescale $k_d$ values to different temperatures via the relation (Montserrat et al., 2017):

$$k_{d2} = k_{d1} \cdot \exp\left(\frac{E_a}{R}\left(\frac{1}{T_2} - \frac{1}{T_1}\right)\right) \qquad (9)$$

Past olivine dissolution studies have typically been conducted at temperatures of 25–125°C (Fig. 4b), from which $E_a$ values of 70.4 kJ mol$^{-1}$ for pH < 5.6 and 60.9 kJ mol$^{-1}$ for pH > 5.6 have been derived (Rimstidt et al., 2012). However, there is a clear lack of rate studies within the environmentally relevant temperature range of 0–25°C, which reflects the natural variation of annual mean temperatures within the seafloor. The few experiments investigating dissolution below 25°C have found markedly lower E$_a$ values of 31–33 kJ mol$^{-1}$ (Hausrath and Brantley, 2010; Niles et al., 2017), but these low-temperature activation energies were derived from experiments conducted in highly acidic conditions (pH < 2; Hausrath and Brantley 2010; Niles et al. 2017). Still, they suggest that the activation energy of 70.4 kJ mol$^{-1}$ at pH < 5.6 used by (Rimstidt et al., 2012) may not hold for the temperature range 0–25°C. Only recently, have studies been conducted at environmentally relevant conditions (pH ~8 and temperature 0–25°C, e.g., Flipkens et al., 2023b; Fuhr et al., 2023, 2024). However, E$_a$ was not reported in these studies, and so the value of E$_a$ is still uncertain under natural conditions, pinpointing an area for further research.

By applying the Arrhenius equation and using an apparent activation energy of 60.9 kJ mol$^{-1}$ (Rimstidt et al., 2012), we find that olivine dissolves about 10 times faster at 25°C compared to at 0°C (Hangx and Spiers, 2009). From a mERW perspective, the geographical location is thus expected to impact the olivine dissolution rate profoundly. Spreading olivine in a polar region (temperature range 0–10°C) compared to application in tropical regions (temperature range 20–35°C) would decrease the dissolution rate by a factor of 2.4–21.0. The sensitivity of $k_d$ to temperature further implies that dissolution rates may vary with the seasons in temperate environments. This potentially temporal variability in the olivine dissolution rate complicates the MRV of mERW, as multiple measurements of olivine dissolution rates would be required at suitable temporal resolution throughout the seasonal cycle, thus increasing the costs of monitoring schemes.

### 3.1.4 Impact of salinity on olivine dissolution

The dissolution rate constant $k_d$ decreases at high salinities (brine, ionic strength > 6 mol kg$^{-1}$ solution), as the activity coefficient of water ($a_{H_2O}$) is lowered when the ionic strength of an aqueous solution increases (Olsen et al., 2015; Prigiobbe et al., 2009). As $a_{H_2O}$ expresses how easily the water can interact with the olivine, a high ionic strength of the solution indirectly affects the dissolution rate by lowering the "effective concentration" of the water needed to react with olivine. When correcting for the change in $a_{H_2O}$, salinity does not noticeably affect the dissolution rate of olivine below an ionic strength of 12 mol kg$^{-1}$ (for reference: open ocean seawater has an ionic strength of ~0.7 mol kg$^{-1}$; Olsen et al. 2015). It should be noted,

however, that most olivine dissolution experiments have been conducted in deionized water. Furthermore, olivine dissolution experiments at seawater salinities have mostly been conducted within the low pH range 1–4 (Fig. 4a). Accordingly, the effect of salinity on olivine dissolution at environmental field conditions (ionic strength 0–0.7 mol kg$^{-1}$, pH 6–8.3) has not been explicitly assessed, but salinity variations are unlikely to have a major effect.

### 3.1.5 Specific surface area

The specific surface area of the olivine grains is a crucial parameter since it is used to normalize $k_d$ under the assumption that dissolution rates are surface-controlled (Brantley et al., 2008). Typically, surface areas are reported as either geometric ($A_{\text{geo}}$) or BET ($A_{\text{BET}}$) surface areas (Brantley et al., 2008; Brunauer et al., 1938; Rimstidt et al., 2012). When using $A_{\text{geo}}$, one assumes that the mineral grain adopts a spherical shape (Brantley et al., 2008). Accounting for spherical geometry, the geometric specific surface area is calculated for a grain size distribution consisting of $n$ classes via (Flipkens et al., 2023b; Rimstidt et al., 2012):

$$A_{\text{geo}} = \sum_{i=1}^{n} \left( \varphi_i \frac{6V_{\text{olivine}}}{M_{\text{olivine}} D_{e_i}} \right) \tag{10}$$

Here, $\varphi_i$ is the forsterite fraction of a certain grain diameter class $i$, $V_{\text{olivine}}$ is the molar volume of forsterite ($4.365 \times 10^{-5}$ m$^3$ mol$^{-1}$), $M_{\text{olivine}}$ is the molar mass of forsterite (140.69 g mol$^{-1}$), and $D_e$ is the equivalent diameter (m) of a grain size interval (Tester et al., 1994):

$$D_e = \frac{D_{\max} - D_{\min}}{\ln\left(D_{\max}/D_{\min}\right)} \tag{11}$$

In Eq. (11), the grain-size distribution is assumed to be constant over the given range. At small particle size intervals, the arithmetic mean of the maximum ($D_{\max}$) and minimum ($D_{\min}$) grain diameter of a particular grain size class will be close to $D_e$. For large intervals, $D_e$ is smaller than the arithmetic mean since smaller particles contribute more to the area than large particles (Rimstidt et al., 2012; Tester et al., 1994).

The BET-based quantity $A_{\text{BET}}$ is obtained by measuring the adsorption of a monolayer of inert gas (Kr, N$_2$) on a sample of grains (Brantley et al., 2008; Brantley and Mellott, 2000). Since the size of a water molecule is similar to that of N$_2$ and Kr, the BET method is considered to be a good proxy for characterizing the interaction between the mineral surface and water (Rimstidt et al., 2012). BET measurements are typically reported for the fresh olivine grains introduced at the start of the experiment, and because of the limited duration of experiments, it is assumed that the specific surface area does not markedly change (e.g. Hänchen et al. 2006; Pokrovsky and Schott 2000; Oelkers 2001b). However, under field conditions, the full dissolution of olivine sand may require 10-500 years, depending on the initial grain size. Accordingly, the evolution of $A_{\text{BET}}$ as the grain dissolves is unknown, which must be considered in future long-term experiments.

The relation between $A_{\text{geo}}$ and $A_{\text{BET}}$ is given by the grain roughness ($R_S$), which is defined as (Brantley et al., 2008; Brantley and Mellott, 2000; Oelkers et al., 2018):

$$R_S = \frac{A_{BET}}{A_{geo}} \qquad (12)$$

Fig. 5a shows the data distribution of $R_S$ values as reported in the literature, which is largely based on Rimstidt et al. (2012)
with a few additions. Following Rimstidt et al. (2012), we excluded data where $R_S$ was larger than 10, as these values likely result from reading or calibration errors and retention of fine particles on the grain surfaces. Additionally, in our data compilation, we omitted the data from Olsen et al. (2015), as they derived a BET value using an empirical rate equation from $A_{geo}$ rather than a direct measurement. The $R_S$ value in our compiled dataset ranges between 2 and 8, with a mean of 5.2 (Fig. 5a). Accordingly, there is considerable variation in grain roughness $R_S$ between the olivine sand used in experiments.
Moreover, there is no strong correlation between $A_{BET}$ and $A_{geo}$ (Fig. 5b). Overall, $R_S$ tends to increase at smaller grain sizes (Strefler et al., 2018), albeit with substantial variability. Overall, the source of this variability remains poorly understood, but it has been shown that the type of mill used to grind olivine source can significantly alter grain roughness $R_S$, providing highly different $A_{BET}$ values for the same $A_{geo}$ (Summers et al., 2005). The lack of a clear empirical relation between $A_{BET}$ and $A_{geo}$ complicates model-based predictions of the dissolution rate in future mERW applications. Therefore, an accurate
characterization of $R_S$ is highly advisable in future mERW experiments and applications. This issue also highlights the importance of a consistent normalization of reported dissolution rate constants $k_d$. Rimstidt et al. (2012) argued that geometric-based normalization of dissolution rates can be useful for practical reasons, as it reduces analysis costs and avoids errors during BET measurements. However, using the actual $A_{BET}$ value is advantageous due to the poor predictability of $R_S$.


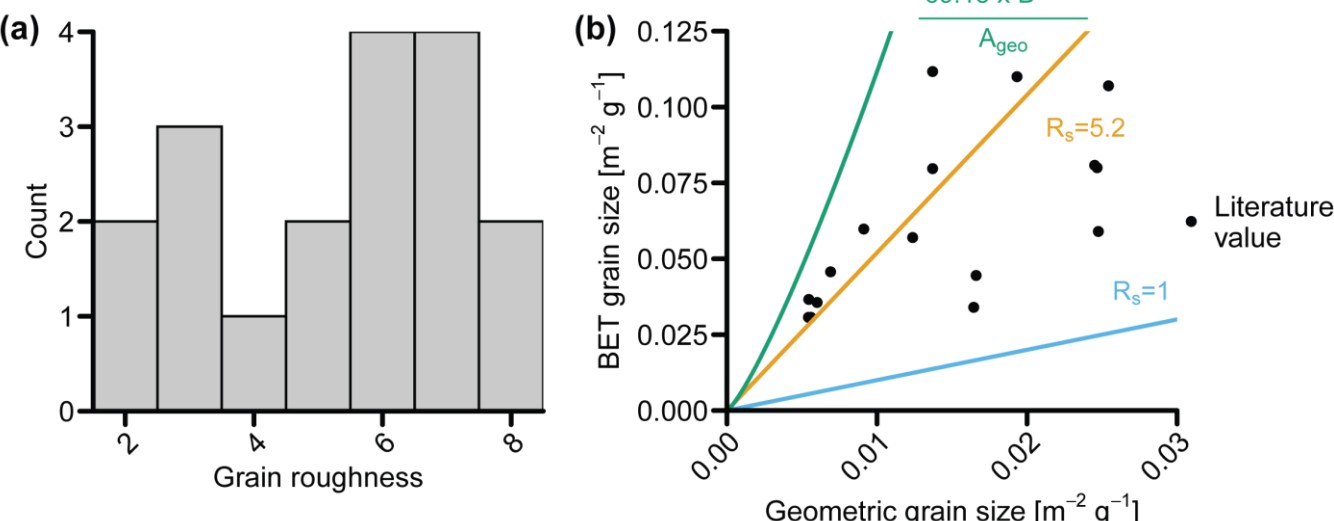

Figure 5. (a) Histogram of grain roughness ($R_S$) values in the compiled literature (supplementary file 2). (b) Relationship between the geometric surface area ($A_{geo}$) and BET surface area ($A_{BET}$). The empirical equation from Strefler et al. (2018) (in green), where D is the grain diameter (µm), corresponds to an average $R_S \sim 12$. $R_S = 5.2$ is the mean grain roughness in the compiled literature, $R_S = 1$ shows the relationship for a perfect sphere.

Furthermore, it should be noted that the link between specific surface area and mineral reactivity can change with time. In the olivine dissolution experiments conducted by Grandstaff (1978), $A_{BET}$ increased substantially over five days (pH = 2.6) as etch pits and cracks formed. These dissolution features have also been found in naturally weathered olivine (Velbel, 2009) as well as laboratory olivine dissolution experiments (e.g. Flipkens et al. 2023b), and hence, the assumption that $A_{BET}$ remains constant in time throughout long-term mERW applications is likely not valid. Interestingly, this large increase in BET was not

accompanied by a concurrent increase in dissolution rate (Grandstaff, 1978), highlighting that there can be an increase in the total surface area while the reactive surface area stays the same. Rimstidt et al., (2012) identified this inconsistency as the major reason for the large data spread found between dissolution experiments (Fig. 4a). Unfortunately, temporal variation in reactive surface area is not easily monitored (Brantley et al., 2008; Oelkers et al., 2018; Rimstidt et al., 2012). Potentially, insight into the evolution of the reactive surface area over the whole weathering trajectory might be attained by looking at

natural deposits of olivine sand with different ages (e.g. the older olivine deposit at Papakōlea beach in Hawaii versus the much younger Tremblet Beach in La Reunion). Comparing $A_{BET}$ and $k_d$ values of freshly mined olivine with those of naturally occurring olivine sand could indicate how olivine weathering rates in mERW applications may be affected over longer time scales.

### 3.1.6 Passivating layers

Passivating layers are any coatings on the olivine surface, biogenic or inorganic, that slow down dissolution. Microbial biofilms growing on olivine grains have been found to weakly inhibit dissolution (Shirokova et al., 2012), and precipitation of secondary minerals (sections 3.2) may also create passivating layers (Béarat et al. 2006; Sissmann et al. 2013). The occurrence of amorphic passivating layers on olivine grains and their effect on dissolution has been discussed in detail elsewhere (Pokrovsky and Schott, 2000; Béarat et al., 2006; Daval et al., 2011; Sissmann et al., 2013; Johnson et al., 2014), and was also extensively

covered in the recent review by Oelkers et al. (2018). Amorphic layers may form as $Mg^{2+}$ is leached from the olivine surface, causing the remaining silica-rich layer to repolymerize and form an amorphous $SiO_2$ coating (Béarat et al., 2006; Oelkers et al., 2018). However, $SiO_2$ coatings are only inhibiting when $Fe^{3+}$ (derived from the fayalite component of the olivine itself) is incorporated into their structure (Oelkers et al., 2018). Recent experiments indicate that microbial and fungal uptake of either Fe (Gerrits et al., 2020, 2021; Lunstrum et al., 2023; Torres et al., 2019), or Fe and Si (Li et al., 2024), can prevent the formation

of these passivating layers and increase olivine weathering rates. However, when no Fe uptake occurs, the effect of microbial biofilms on dissolution is less conclusive but appears to be slightly inhibiting (Shirokova et al., 2012). Future studies should consider *in situ* microbial effects on olivine dissolution, which is likely relevant for weathering in natural systems.

In their olivine dissolution experiments in seawater, Montserrat et al. (2017) observed a decreased Mg-to-Si atomic ratio on the surface of reacted forsterite compared to the initial substrate, indicating that a Mg-leached layer had formed as was

proposed by Hellmann et al. (2012) and Maher et al. (2016). In contrast, Fuhr et al. (2022) found no such Mg-depletion, postulating that the high Mg concentrations in their artificial seawater prevented depletion at the olivine surface. Flipkens et al. (2023) mimicked the wave action in the coastal zone by continuously rotating olivine sand to induce grain collision,

hypothesizing that grain abrasion could decrease the formation of passivating layers. These experiments revealed that physical agitation increases the olivine dissolution rate by a factor of 8–19 compared to stagnant conditions. However, Flipkens et al. (2023) only found slight variations in the Mg-to-Si atomic ratio on the olivine grain surface between treatments (with large variations on the same grain), indicating that the formation of Mg-depleted layers was minimal in both the stagnant and high rotation treatments. Flipkens et al. (2023) hence attributed most of the dissolution rate enhancement to water flushing rather than physical abrasion of the grain surface. Particle abrasion of olivine has also been found to strongly influence the rate of mineral carbonation at high temperature and pressure, where the formation of passivating layers is likely more pervasive compared to mERW (Béarat et al., 2006).

Until now, there is little knowledge on passive layer formation on olivine grains under *in situ* marine conditions relevant to mERW. Previous experiments reporting on passivating layer formation have been typically conducted in freshwater at elevated $CO_2$ pressure (135–250 bar) and temperature (60–185°C) (Béarat et al., 2006; Daval et al., 2011; Johnson et al., 2014; Maher et al., 2016; Sissmann et al., 2013). Consequently, the possibility of passive layer formation in coastal sediment conditions warrants further research. Several processes may potentially counteract such layer formation during mERW application. In the bedload scenario, olivine particles will be subjected to the action of waves and currents, possibly removing passivating layers (Flipkens et al., 2023b). Moreover, when olivine is applied in cohesive sediment, the particles may be ingested by infauna, which could prevent passivating layer formation (Meysman and Montserrat, 2017). Although initial studies suggest that substantial formation of passivating layers seems unlikely (Bach, 2024; Flipkens et al., 2023b; Montserrat et al., 2017), the formation of a thin cation-depleted layer cannot be completely ruled out. Future work should investigate the precise conditions under which these layers form and how they impact weathering rates (Montserrat et al., 2017; Oelkers et al., 2018; Palandri and Kharaka, 2004; Pokrovsky and Schott, 2000), as well as further quantify the effect of grain abrasion on passivating layer formation (Flipkens et al., 2023b). Likewise, the prevalence and effect of biofilm formation on olivine dissolution rates need to be addressed in future mERW studies.

## 3.2 Secondary mineral formation

Chemical weathering of olivine is commonly described as a reaction where pure forsterite completely dissociates into its constituent ions $Mg^{2+}$ and $SiO_4^{4-}$, thus generating 4 moles of alkalinity per mole of olivine dissolved (Table 3) (Meysman and Montserrat, 2017; Oelkers et al., 2018; Schuiling and Krijgsman, 2006). However, the alkalinity production can be decreased by the formation of secondary minerals via (i) incomplete olivine dissolution and (ii) precipitation of dissolved weathering products (Bach, 2024; Flipkens et al., 2023b; Fuhr et al., 2022; Griffioen, 2017; Meysman and Montserrat, 2017; Montserrat et al., 2017; Rigopoulos et al., 2018) (Table 3). During incomplete dissolution, alteration products such as serpentine or assemblages of clay minerals (e.g. iddingsite and smectites) are formed directly from the olivine, resulting in a lowered (or null) alkalinity production. Olivine that has dissolved completely can in turn contribute to the build-up of dissolution products in the pore water and promote the precipitation of minerals (e.g., carbonates, iron minerals, clays), which consumes alkalinity

generated from olivine weathering. The relative reduction of alkalinity generation through secondary mineral formation can be described by the alkalinity production factor $\gamma_{A_T}$, which ranges from 0–1 (Table 3). Both the formation of olivine alteration products and the precipitation of minerals from dissolution products are regulated by kinetics, the availability of nucleation sites, and the saturation state of a particular mineral, with the latter being strongly affected by the water exchange rate. These

reactions are not exclusive and can occur in tandem, meaning that the overall value of $\gamma_{A_T}$ should represent the weighted contribution of the relevant reactions to the alkalinity loss. Recently, several experiments have been conducted on olivine dissolution in both artificial seawater (e.g. Montserrat et al. 2017; Rigopoulos et al. 2018; Fuhr et al. 2022) and natural seawater (Flipkens et al., 2023b; Fuhr et al., 2024; Montserrat et al., 2017). These studies provide a first insight into the secondary minerals that can form during enhanced olivine weathering. However, it is unclear to what extent the reaction conditions (e.g.

pH and chemical environment) utilized in these laboratory experiments are relevant for olivine dissolution in sediments, and studies conducted on olivine weathering in sediment have indicated that secondary mineral formation is possible, but no direct quantification of precipitates has been performed (Bach, 2024; Fuhr et al., 2023, 2024). In the sections below, we discuss the likelihood of formation of different secondary minerals during mERW with olivine.

**Table 3. Reactions that can affect the alkalinity generation during mERW with olivine, where $\gamma_{AT}$ is the fraction of alkalinity generated relative to complete olivine dissolution (value 0–1). For $\gamma_{AT}$ calculations, silica was assumed to be the limiting factor sourced exclusively from olivine. For example the precipitation of 1 mol sepiolite corresponds with the dissolution of 6 moles of olivine, producing 24 mol alkalinity. Sepiolite formation consumes 8 moles alkalinity resulting in $\gamma_{AT}$ =0.67. [a]Brucite is assume dto dissolve further increasing $\gamma_{AT}$ to 0.25. [b]No alkalinity is produced, but the reaction leads to the formation of magnesium carbonate,**

**directly sequestering 0.5 mol DIC per mol forsterite. [c]Assuming 0.5 moles of Al are incorporated per 3.5 moles Si. [d]The reaction is unbalanced since iddingsite does not consist of a single phase and its composition varies. [e]The $Fe^{2+}$ stems from fayalite, as olivine typically comprises of 6–20 mol% Fe. [f]Fayalite weathering and sulfate reduction each produce 2 moles of alkalinity per mole of iron ($Fe^{2+}$) and $SO_4^{2-}$, respectively. When both $Fe^{2+}$ and sulfide ($H_2S$) precipitate to form iron sulfide (FeS), 2 moles alkalinity are consumed. Thus, the net result is that 2 moles of alkalinity produced by fayalite weathering are maintained. [g]When calcium**

**carbonate (CaCO3) is formed, 1 mol DIC is sequestered directly within the mineral. Therefore, if all alkalinity from olivine dissolution is consumed in carbonate precipitation, half of the $CO_2$ is sequestered compared to a situation without carbonate precipitation.**

| Reaction | Mechanism | $\gamma_{A_T}$ |
|---|---|---|
| Complete olivine dissolution[1] | $Mg_2SiO_4 + 4H_2O \rightarrow 4OH^- + 2Mg^{2+} + H_4SiO_4$ | 1 |
| Serpentinization[2,3] | $3Mg_2SiO_4 + 4H_2O + SiO_2 \rightarrow 2Mg_3Si_2O_5(OH)_4$ | 0[a] |
| | $2Mg_2SiO_4 + 3H_2O \rightarrow Mg_3Si_2O_5(OH)_4 + Mg^{2+} + 2OH^-$ | 0.25 |
| | $2Mg_2SiO_4 + 2H_2O + CO_2 \rightarrow Mg_3Si_2O_5(OH)_4 + MgCO_3$ | 0[b] |
| Iddingsitization[c,2,5] | $(Mg,Fe)_2SiO_4 \rightarrow MgO\bullet Fe_2O_3\bullet 3SiO_2\bullet 4H_2O$ | 0 |
| Saponite formation[5] | $3.5Mg_2SiO_4 + 0.25Ca^{2+} + 0.5Al^{3+} + (4+n)H_2O \rightarrow$ $4Mg^{2+} + 6OH^- + Ca_{0.25}Mg_3Al_{0.5}Si_{3.5}O_{10}(OH)_2 \cdot nH_2O$ | 0.43[c] |
| Sepiolite formation[3,6] | $6Mg_2SiO_4 + 15H_2O \rightarrow 8Mg^{2+} + 16OH^- + Mg_4Si_6O_{15}(OH)_2 \cdot 6H_2O$ | 0.67 |

| Talc formation[3] | $4Mg_2SiO_4 + 6H_2O \rightarrow 5Mg^{2+} + 10OH^- + Mg_3Si_4O_{10}(OH)_2$ | 0.63 |
|---|---|---|
| Iron oxide precipitation[d] | $Fe^{2+} + \frac{1}{4}O_2 + 2OH^- + \frac{1}{2}H_2O \rightarrow Fe(OH)_3$ | 0 |
| Sulfate reduction + iron sulfide precipitation[e] | $2CH_2O + SO_4^{2-} \rightarrow 2CO_2 + H_2S + 2OH^-$  $Fe^{2+} + H_2S + 2OH^- \rightarrow FeS + 2H_2O$ | 1[f] |
| Carbonate precipitation[3] | $Ca^{2+} + 2HCO_3^- \rightarrow CaCO_3 + CO_2 + H_2O$ | 0[g] |

[1](Meysman and Montserrat, 2017), [2](Griffioen, 2017; Deer et al., 2013), [3](Flipkens et al., 2023b) [4](Edwards, 1938), [5](Wilson, 2004), [6](Isson and Planavsky, 2018)


### 3.2.1 Formation of clay minerals

Clay minerals can form either through direct alteration of the olivine or through authigenic precipitation of dissolution products, also known as reverse weathering (Deer et al., 2013; Delvigne et al., 1979; Isson and Planavsky, 2018). The saturation state of the system, which is primarily driven by the water exchange, is critical for clay mineral formation. When rock-to-fluid ratios are low, $Mg^{2+}$ and $Fe^{2+}$ from olivine dissolution escape into solution (Delvigne et al., 1979; Wilson 2004). When rock-to-fluid ratios are high, dissolution products are concentrated, and clay formation is promoted (Delvigne et al., 1979; Gaudin et al., 2018).

Serpentinization is an olivine alteration process in which forsterite is transformed into the clay serpentine, sometimes in parallel with the formation of brucite ($Mg(OH)_2$) or magnesite ($MgCO_3$) (Table 3) (Deer et al., 2013; Griffioen, 2017). No alkalinity is formed during serpentinization $\left(\gamma_{A_T} = 0\right)$, but precipitation of magnesite leads to direct sequestration of $CO_2$ in the mineral. When the brucite formed during serpentinization dissolves further, $\gamma_{A_T}$ increases accordingly to 0.25. In marine environments, serpentine is typically formed over a wide temperature range (50–600°C) in high-temperature hydrothermal systems, subduction zones, and mid-ocean ridges (Alt et al., 2012, 2013; Schwarzenbach et al., 2012). Serpentinization has also been found to occur at lower temperatures (from 200°C down to ~10°C) in hydrothermal deposits with high sulfide content, such as the Lost City hydrothermal field (Mid-Atlantic ridge) (Alt et al., 2012; Mével, 2003; Neubeck et al., 2011).

Saponite (a smectite mineral, formerly classified as bowlingite) and iddingsite are often identified as alteration products of olivine, although saponite can also be formed through precipitation (Deer et al., 2013; Dehouck et al., 2016; Delvigne et al., 1979; Gaudin et al., 2018; Isson and Planavsky, 2018) . The formation of saponite reduces the alkalinity production to 1.7 moles alkalinity mol[-1] olivine $\left(\gamma_{A_T} = 0.43\right)$ and the formation of iddingsite does not lead to any alkalinity production $\left(\gamma_{A_T} = 0\right)$ (Isson and Planavsky, 2018). The relative abundance of saponite and iddingsite is dependent on the oxidation state of Fe, with saponite being formed under non-oxidative conditions and iddingsite under oxidative conditions (Deer et al., 2013; Smith et al., 1987). Sepiolite has also been suggested to precipitate from olivine dissolution products (Griffioen, 2017) and reduces the alkalinity production to 2.68 moles alkalinity mol[-1] olivine $\left(\gamma_{A_T} = 0.67\right)$ (Table 3). Sepiolite formation has been observed in

low-temperature systems, such as near hydrothermal vents, in marine and lacustrine sediments, and in volcanic deposits (Mulders and Oelkers, 2020; Wollast et al., 1968). Sepiolite formation requires a high pH $> 8$ (Baldermann et al., 2018; Wollast et al., 1968) and elevated concentrations of dissolved Si and $Mg^{2+}$ (Baldermann et al., 2018; Tosca and Masterson, 2014). Since seawater contains high concentrations of $Mg^{2+}$ (0.05 mol kg$^{-1}$; Johnson et al., 1992), dissolved Si is generally the limiting element for sepiolite formation in marine environments (Baldermann et al., 2018). However, even at a steady supply of dissolved Si (e.g., through silicate weathering or dissolution of biogenic opal) and at a high pH, precipitation of sepiolite remains slow (~$10^{-12}$ mol s$^{-1}$) at ambient coastal seawater temperatures (Baldermann et al., 2018). In marine conditions, sepiolite formation is further impeded by the presence of sodium ions ($Na^+$), which affects the speciation of dissolved silicate, and by the presence of sulfate ions ($SO_4^{2-}$), which lowers the availability of free $Mg^{2+}$ through the formation of $MgSO_4$ complexes (Baldermann et al., 2018; Tosca and Masterson, 2014).

Up until now, serpentine has not been found to form in olivine dissolution experiments using oxic seawater (Flipkens et al., 2023b; Fuhr et al., 2022; Montserrat et al., 2017; Rigopoulos et al., 2018), although both Fuhr et al. (2022) and Flipkens et al. (2023) calculated that serpentine was oversaturated throughout most of their experiments. However, serpentinization is unlikely during mERW as the process likely is negligeable at the temperature range of coastal systems (0–35°C). Serpentinization with magnesite production is particularly unlikely as this mineral typically does not form under ambient marine conditions (section 3.2.3) (Griffioen, 2017). Fuhr et al. (2022) and Flipkens et al. (2023) also advanced that talc ($Mg_3Si_4O_{10}(OH)_2$) formation is possible. However, talc formation from olivine proceeds via (or co-occurs with) serpentine, and these reactions require high pressure and temperature (200–650°C; Bucher and Grapes 2011; Deer et al. 2013). As such, the formation of talc is unlikely in coastal settings. Little experimental data exists on the formation of saponite and iddingsite during olivine dissolution. Even though the formation of saponite and iddingsite during olivine dissolution has been historically well-documented for soils (Baker and Haggerty, 1967; Brown and Stephen, 1959; Edwards, 1938; Eggleton, 1984; Sherman and Uehara, 1956; Smith et al., 1987; Sun, 1957; Wilshire, 1958), the process has not been investigated in seawater conditions. Model calculations have also shown sepiolite oversaturation in studies on olivine dissolution in seawater (Flipkens et al., 2023b; Fuhr et al., 2022; Rigopoulos et al., 2018), yet documentation of sepiolite formation remains inconclusive. Sepiolite has only been directly observed once (Rigopoulos et al., 2018), while another study found a mix of silicate-bearing phases precipitated on weathered olivine grains, potentially including sepiolite (Fuhr et al., 2022). In contrast, Montserrat et al. (2017) and Flipkens et al. (2023) found no phyllosilicate precipitates. Oelkers et al. (2018) suggested that sepiolite formation in the experiments by Rigopoulos et al. (2018) could have been provoked by the buildup of $Mg^{2+}$ in the supernatant solution. However, this is unlikely due to the high background concentration of $Mg^{2+}$ in seawater. Instead, a rapid pH increase (from 8 to >8.6) caused by rock flour addition may have caused the observed precipitation, since a pH increase from 8 to 9 can increase the sepiolite growth rates by an order of magnitude (Baldermann et al., 2018). Overall, due to the generally low pH of pore water and low ambient temperatures in coastal environments, sepiolite formation is unlikely during mERW application, but more targeted studies are needed to confirm this. While clay formation should be targeted by future mERW studies, they are

unlikely to find well-crystalized clays. Instead, authigenic clay minerals are likely to appear as "poorly crystalline gels", which act as precursor complexes (Tosca and Masterson, 2014), similar to the observations by Rigopoulos et al. (2018).

### 3.2.2 Iron mineral formation

Dissolution of fayalite releases $Fe^{2+}$ to the pore water, which in oxic environments spontaneously precipitates as iron (hydr)oxides during aerobic oxidation (Table 3) (Griffioen, 2017). Iron oxidation consumes 2 moles of alkalinity per mole of $Fe^{2+}$, equaling the alkalinity produced during fayalite dissolution ($\gamma_{A_T} = 0$). Assuming complete dissolution of olivine followed by precipitation of the 6–20 mol% Fe in olivine (Ackerman et al., 2009; Deer et al., 2013; Harben and Smith, 2006; Rehfeldt et al., 2007; Su et al., 2016) as iron oxides, 3.2–3.76 moles of alkalinity are produced per mol of olivine. The formation of iron

(hydr)oxides has been confirmed in olivine dissolution experiments in seawater (Fuhr et al., 2022; Rigopoulos et al., 2018). Precipitation of iron (hydr)oxides is also expected from a thermodynamic perspective, as these experiments were conducted under well-oxygenated conditions. During mERW applications, well-flushed and well-oxygenated sediments are targeted to avoid porewater saturation effects, so the alkalinity reduction due to Fe oxidation from fayalite needs consideration.

If olivine particles are buried in deeper, anoxic sediment layers, the $Fe^{2+}$ released from fayalite dissolution can react with $H_2S$

and form iron sulfides ($FeS_x$, Table 3). Like iron oxide formation, the precipitation of the $FeS_x$ consumes the alkalinity formed through fayalite dissolution. However, the alkalinity formed via sulfate reduction is preserved when $H_2S$ is prevented from reoxidizing through trapping in the sediment (Hu and Cai, 2011; Middelburg et al., 2020), causing a net alkalinity production of 4 mol per mol olivine and thus giving a $\gamma_{A_T}$ of 1. However, the fate of any $FeS_x$ minerals produced must be closely scrutinized, as it will determine the permanence of the alkalinity produced. If the $FeS_x$ comes into contact with $O_2$ (e.g. through

bioirrigation or physical advection), they will be reoxidized, and the alkalinity produced during their precipitation will be consumed again (Schippers and Jørgensen, 2002). The long-term fate of ferrous iron released during mERW trials is not well constrained, and will most likely be strongly dependent on the application scenario (cohesive versus permeable). It is hence an important point of attention in future mesocosm experiments and field trials.

### 3.2.3 Stimulated precipitation and inhibited dissolution of carbonates

The release of alkalinity from olivine weathering increases the saturation state of carbonates, which can lead to precipitation of these minerals in the sediment or the water column. Carbonate precipitation consumes 1 mol of dissolved inorganic carbon (DIC) and 2 moles of alkalinity (Table 3), hence leading to the outgassing of $CO_2$ to the atmosphere (Wolf-Gladrow et al., 2007). In theory, different carbonate minerals can form. Given the release of $Mg^{2+}$ and $Fe^{2+}$ during olivine dissolution, the formation of magnesite ($MgCO_3$) and siderite ($FeCO_3$) needs consideration. However, $MgCO_3$ is unlikely to form during

mERW in coastal sediments, as it typically only precipitates at elevated $CO_2$ pressure or temperature (60–100°C) (Griffioen, 2017; Saldi et al., 2009). Likewise, the formation of $FeCO_3$ requires particular environmental conditions, including anoxic but hydrogen sulfide ($H_2S$) free conditions with high concentrations of $Fe^{2+}$ and a narrow pH window between 6.0 and 7.2 (Lin et al., 2020). Therefore, the most likely carbonates to precipitate are calcium carbonates ($CaCO_3$). Coastal waters are

oversaturated with respect to $CaCO_3$ (Morse et al., 2007), and so there is concern that the addition of alkalinity through OAE could induce precipitation of the mineral (Hartmann et al., 2023; Moras et al., 2022). The saturation state with respect to $CaCO_3$ is highly dependent on the type of sediment. In more permeable, sandy sediments, oxic respiration processes lower the pH and can result in undersaturation and dissolution of $CaCO_3$ (Milliman and Droxler, 1996; Morse and Mackenzie, 1990; Rao et al., 2014). While dissolution of $CaCO_3$ may be considerable in cohesive sediments (e.g., Rao et al., 2014), this process can be counteracted by anoxic respiration processes that produce alkalinity and elevate the saturation state of $CaCO_3$ (Berner, 1984; Turchyn et al., 2021). However, a high saturation state does not necessarily lead to $CaCO_3$ precipitation due to inhibitors such as phosphate and organic matter in the sediment (Morse et al., 2007; Turchyn et al., 2021). Flushing of the sediment through advection or bioirrigation also prevents alkalinity from building up (Rao et al., 2012).

Olivine dissolution experiments in seawater have generated mixed and conflicting results regarding the importance of secondary $CaCO_3$ formation. In some experiments, $CaCO_3$ formation was observed (Fuhr et al., 2022; Rigopoulos et al., 2018), while other experiments did not show any precipitation (Bach, 2024; Flipkens et al., 2023b; Montserrat et al., 2017). One factor causing $CaCO_3$ precipitation could be that the seawater was isolated from the atmosphere in some experiments (Fuhr et al., 2022; Rigopoulos et al., 2018), whereas no precipitation was observed in experiments where gas exchange was allowed (Flipkens et al., 2023b; Montserrat et al., 2017). The latter ensures that $CO_2$ can move into solution as alkalinity is produced, thus preventing a pH rise and resulting in $CaCO_3$ oversaturation. Another explanation for the varying results could be the use of different types of seawater in the experiments. The presence of phosphate may have inhibited $CaCO_3$ formation in experiments with natural seawater (Bach, 2024; Flipkens et al., 2023b; Montserrat et al., 2017), in contrast to experiments conducted with phosphate-free artificial seawater (Fuhr et al., 2022; Rigopoulos et al., 2018). As predicted by thermodynamic modeling (Griffioen, 2017), no magnesite ($MgCO_3$) has been detected in olivine dissolution experiments in seawater (Flipkens et al., 2023b; Fuhr et al., 2022; Montserrat et al., 2017; Rigopoulos et al., 2018). Siderite ($FeCO_3$) formation was not explicitly investigated in the experiments, but as noted above, this process is unlikely based on theoretical grounds.

Although mERW might not cause precipitation of $CaCO_3$ in the sediment, it could impede natural alkalinity production by inhibiting the dissolution of $CaCO_3$ already present in the sediment (Bach, 2024). As such, the alkalinity sourced from carbonate dissolution is substituted with alkalinity from olivine dissolution, so the alkalinity generated by mERW is no longer fully additional to the alkalinity efflux before mERW application. While the additionality problem may imply that mERW does not increase the sedimentary alkalinity release, the $CO_2$ sequestration potential still increases when alkalinity originates from olivine, as $CaCO_3$ dissolution results in a concomitant release of alkalinity and DIC, while no DIC is produced during olivine dissolution (Bach, 2024). The coastal sediments in which $CaCO_3$ dissolution is high are those where the pH is lowered due to oxic mineralization of organic matter, thus overlapping with the optimal environmental conditions for olivine dissolution. The additionality problem may thus be an issue for mERW in both the permeable and cohesive sediment application scenarios (Fig. 2c–d), in which the dissolution of olivine dissolution occurs at the location of carbonate dissolution. If areas with permeable or cohesive sediments are targeted for mERW, it should first be confirmed that they are not strong sources of alkalinity from natural $CaCO_3$ dissolution. However, the additionality problem should not be an immediate issue

for the bedload application (Fig. 2b), as the olivine is weathering in seawater that is already oversaturated with respect to $CaCO_3$, and there is also an immediate dilution of the alkalinity.

In conclusion, the rate at which mERW applications influence natural alkalinity release from sediments imposes a major uncertainty on $\gamma_{A_T}$. The additionality problem could have profound implications for the location chosen for mERW application and the economics and MRV of a mERW project, as the net alkalinity generation must be quantified rather than the total sedimentary alkalinity release. Hence, future mERW experiments need to quantify the natural alkalinity production in control sediments and determine to what extent this process is affected by mineral addition.

mERW targets a slow release of alkalinity from the sediment to the water column, however, other OAE approaches, such as ocean liming and electrochemistry, target a fast, immediate release of alkalinity (Fig. 1). These fast-release approaches can result in substantial alkalinity increases at the local scale. In recent liming experiments, the addition of ~250 µmol kg$^{-1}$ alkalinity on top of a seawater baseline of 2400 µmol kg$^{-1}$ (a ~10% increase, equivalent to an aragonite saturation index of ~5) was delineated as a safe threshold to avoid $CaCO_3$ precipitation in the water (Moras et al., 2022). In contrast, the addition of

500 µmol kg$^{-1}$ of alkalinity led to "runaway precipitation", a process where $CaCO_3$ precipitation continued until the aragonite saturation reached 1.8–2.0 and the alkalinity concentration had been reduced below the initial seawater values (Hartmann et al., 2023; Moras et al., 2022). However, the risk of runaway precipitation is small when using olivine for mERW. When applying olivine grains of 10 µm at a loading of 15 kg olivine m$^{-2}$ in a very shallow system (1 meter of overlying water, equivalent to the 15 g l$^{-1}$ used by, Montserrat et al., 2017), the alkalinity concentration in the overlying water increases by ~170

µmol kg$^{-1}$ seawater over the course of a day (supplementary file 1.3). This alkalinity increase is well below what has been deemed safe in past run-away experiments (Moras et al., 2022). In reality, this alkalinity would not continue to build up in the overlying water but would be diluted, further minimizing the risk of immediate $CaCO_3$ precipitation in the overlying water. Finally, alkalinity generation through olivine addition could stimulate the growth of calcifying algae and lead to loss of alkalinity through increased biogenic $CaCO_3$ precipitation in the water column. However, while the addition of olivine appears

to stimulate the growth of phytoplankton in general, there is a much more pronounced effect on silicifying algae (Li et al., 2024), which out-compete other phytoplankton such as calcifying coccolithophores (Bach et al., 2019; Li et al., 2024). These initial studies thus indicate that the risk of substantial alkalinity loss in the water column due to increased biogenic calcification is small.

### 3.2.4 Implications for mERW with olivine

Apart from the precipitation of Fe oxides following fayalite dissolution, experiments on olivine dissolution in seawater have so far provided little evidence of extensive secondary mineral formation via incomplete olivine dissolution or authigenic precipitation. In one instance, small amounts of sepiolite formation were reported (Rigopoulos et al., 2018), whereas no serpentinization has been observed (Montserrat et al., 2017; Rigopoulos et al., 2018; Fuhr et al., 2022; Flipkens et al., 2023b). Most mERW experiments have been conducted in oxygenated seawater and are not fully representative of sedimentary

conditions (e.g. in terms of $O_2$ and pH), and studies in sediments under hypoxic-anoxic bottom water have not directly

measured the formation of secondary minerals (Fuhr et al., 2024). Most olivine dissolution experiments have also been conducted in closed systems, where reaction products are allowed to build up (e.g. Montserrat et al., 2017; Rigopoulos et al., 2018; Fuhr et al., 2022). Under mERW application, the weathering will take place in an open environment. Reaction products from olivine weathering will be released into the pore water, which can then be flushed either through physical advection

(wave or current action) or through bioirrigation (Meysman and Montserrat, 2017). It should be noted that Bach (2024) and Fuhr et al. (2023, 2024) included sediment in their ERW experiments; however, they did not specifically quantify secondary mineral formation. While the physiochemical conditions in coastal sediments seem unfavourable for the formation of serpentine, the formation of iddingsite and smectites (most notably saponite) could be relevant depending on flushing conditions, and their formation during mERW warrants further research. Similarly, the effect of olivine dissolution on the

inhibition of natural carbonate dissolution may be important in cohesive and permeable sediments and should be investigated in more detail.

## 3.3 The alkalinity transfer efficiency and $CO_2$ sequestration efficiency

The final step in the mERW process represents the transfer of alkalinity to the surface water and subsequent equilibration of

$CO_2$ at the air-sea interface (Fig. 2a). The upward transfer of alkalinity to the sea surface occurs on a time scale of weeks depending on the water depth (He and Tyka, 2023; Jones et al., 2014; Zhou et al., 2024), and the air-sea $CO_2$ exchangetakes place on a time scale of months up to a year, with longer timescales associated with deep mixed layers (He and Tyka, 2023; Jones et al., 2014; Zhou et al., 2024). The proportion of alkalinity that reaches the surface water and equilibrates with the atmosphere is given by the alkalinity transfer efficiency $\eta_{A_T}$, a time-dependent factor that varies between 0 and 1. Model

simulations from He and Tyka (2023) and (Zhou et al., 2024) demonstrated that, when alkalinity is added to surface waters, many coastal systems reach $\eta_{A_T}$ values of 0.8–1 after 3–4 years, while lower $\eta_{A_T}$ values (~0.44–0.75) were associated with specific coastal systems that display deep-water formation. High values for $\eta_{A_T}$ were also associated with highly stratified systems, since the surface-released alkalinity remains in contact with the atmosphere for longer (He and Tyka, 2023; Zhou et al., 2024). However, these model results cannot be directly applied to mERW, since alkalinity is released from the sediment,

and not directly within the surface water, and so high stratification would prevent sediment-borne alkalinity of reaching the surface. Generally, the $\eta_{A_T}$ factor demonstrates the importance of deploying mERW in areas without downwelling into the deeper ocean and with sufficient water mixing, so that bottom water alkalinity can reach the surface waters to get sufficient contact with the atmosphere. As most mERW modelling studies assume (near) instantaneous alkalinity release into the surface water (e.g., Feng et al., 2017; Hauck et al., 2016; Köhler et al., 2013), the alkalinity transfer efficiency and its effect on the

$CO_2$ sequestration should be accounted for in further studies.

The amount of $CO_2$ that will subsequently be captured as the surface water and atmosphere equilibrate is given by the $CO_2$ sequestration efficiency $\rho_{CO_2}$, which is a thermodynamic quantity defined as (Zeebe and Wolf-Gladrow, 2001):

$$\rho_{CO_2} = \left( \frac{\partial DIC}{\partial A_T} \right)_{f_{CO_2}} \tag{13}$$

This partial derivative represents the change in the concentration of DIC upon the addition of one mole of alkalinity ($A_T$) to the seawater, evaluated at a constant fugacity (or partial pressure) of $CO_2$ ( $f_{CO_2}$ ) (Wolf-Gladrow et al., 2007; Bertagni and Porporato, 2022). The value of $\rho_{CO_2}$ can be calculated using dedicated software packages for seawater carbonate chemistry like AquaEnv (Hofmann et al., 2009) or CO2sys (Xu et al., 2017).

The magnitude of $\rho_{CO_2}$ depends on the local seawater chemistry, temperature, and salinity, causing $\rho_{CO_2}$ to vary between different coastal systems (Middelburg et al., 2020; Bertagni and Porporato, 2022). Across the global ocean, $\rho_{CO_2}$ ranges from ~0.75–0.95 with an average value of 0.84 (Bertagni and Porporato, 2022; Schulz et al., 2023). Figure 6a displays the predicted DIC as a function of alkalinity for a set of endmember conditions for salinity and temperature relevant to coastal systems. This graph illustrates the key mechanism underlying OAE: an increase in the alkalinity of seawater leads to an increase in DIC after equilibration with the atmosphere. The slope of the lines depict $\rho_{CO_2}$ (Bertagni and Porporato, 2022; Schulz et al., 2023), which depends on temperature, salinity, and local seawater chemistry. Colder and less saline waters can store more $CO_2$ (Zeebe and Wolf-Gladrow, 2001) and are characterized by a higher $\rho_{CO_2}$ (Fig. 6a). Figure 6b–d illustrate the impact of local seawater chemistry (i.e., as specified by pH and $f_{CO_2}$ ), temperature, and salinity on $\rho_{CO_2}$. As a reference condition, we use pH = 8.07 (all pH values presented are on the total pH scale), temperature = 15°C, salinity = 35, and current $CO_2$ levels ( $f_{CO_2}$ = 420 ppm).

The surface ocean pH between 60°N to 60°S varies between 8.0 and 8.25, with a global average of $8.07 \pm 0.02$ (Jiang et al., 2019). In coastal areas, riverine input can cause larger variations in the pH, since the pH of rivers typically ranges between 6 and 8 (Mackenzie and Lerman, 2006). As shown in Fig. 6b, the pH exerts a strong influence on $\rho_{CO_2}$ when moving across the entire pH scale from 4 to 12. At pH <5, $\rho_{CO_2}$ effectively becomes zero since the addition of alkalinity here leads to a consumption of $H^+$ rather than the production of $HCO_3^-$ or $CO_3^{2-}$, and hence there is no scavenging of $CO_2$ (Bertagni and Porporato, 2022; Hofmann et al., 2008). At higher pH, $\rho_{CO_2}$ increases to a maximum value of 0.97 at pH = 6.7, before decreasing again beyond pH ~8 and stabilizing at $\rho_{CO_2} \approx 0.5$ at pH > 10. Note that the value of $\rho_{CO_2} = 1$ is never reached in seawater due to the presence of the borate buffer (Bertagni and Porporato, 2022). When considering the typical current ocean seawater pH range of 8.0–8.25, the $\rho_{CO_2}$ variation is minimal with values of 0.83-0.88 (Fig. 6b).

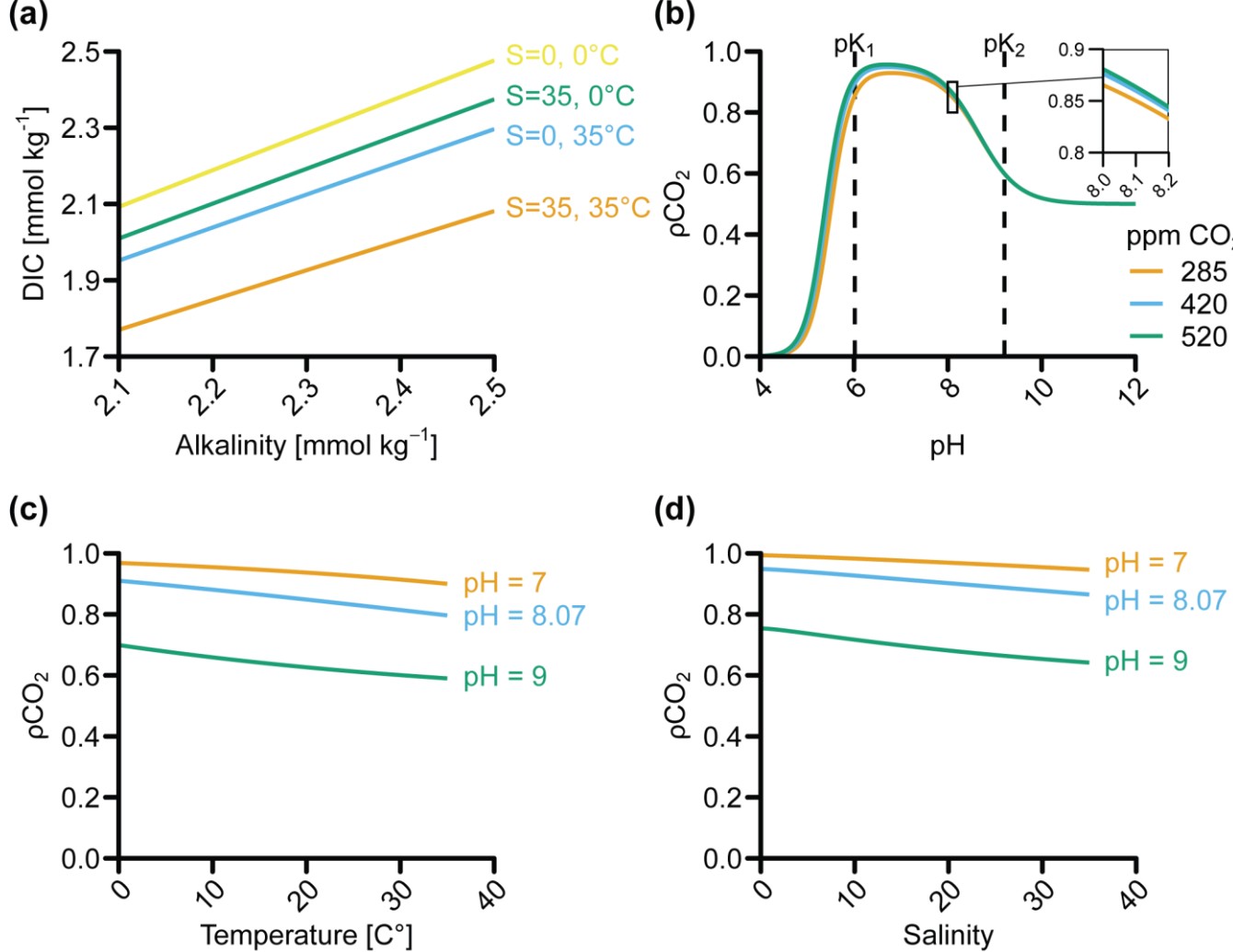

**Figure 6.** The CO₂ sequestration efficiency ($\rho_{CO_2}$) as a function of environmental parameters (unless otherwise stated pH = 8.07, temperature = 15 °C, salinity = 35, $f_{CO_2}$ = 420 ppm). (a) Predicted DIC as a function of alkalinity for a set of endmember temperature and salinity values. (b) $\rho_{CO_2}$ as a function of pH at different levels of atmospheric CO₂ (constant temperature and salinity), with pK values from Lueker et al. (2000). Typical seawater pH is shown in magnification. (c) $\rho_{CO_2}$ as a function of temperature at different pH levels (constant salinity and $f_{CO_2}$).

Temperature and salinity influence $\rho_{CO_2}$ through the stoichiometric equilibrium constants ($K^*$) of the carbonate system: $\rho_{CO_2}$ decreases with increasing temperature (Fig. 6c) and increasing salinity (Fig. 6d). The temperature effect is in large part due to the presence of a borate buffer in seawater, which also contributes to alkalinity. An increase in temperature decreases the solubility of CO₂ without affecting the borate concentration, leading to a decreasing ratio between DIC and borate (Bertagni and Porporato, 2022). Across the ocean, the temperature varies between 0 and 35°C (Sarmiento and Gruber, 2006), which results in a $\rho_{CO_2}$ range from 0.80 to 0.91 (Fig. 6c).

The effect of salinity on $\rho_{CO_2}$ is driven by ion activity. When the salinity (and hence the number of ions in solution) increases, the ion activity decreases. Since the activities of different ions are not affected by the same magnitude, the ratio of their activity coefficients changes. For the carbonate system, the activity coefficient of the bivalent $CO_3^{2-}$ decreases more rapidly with increasing salinity than those of the monovalent ions $HCO_3^-$ and $H^+$. A decrease in salinity thus shifts the entire $\rho_{CO_2}$ curve to the left (Fig. 6a). The impact of salinity on $\rho_{CO_2}$ in the open ocean is minimal, as the variation in surface salinity is in the

range 33–37 (Sarmiento and Gruber, 2006), giving a $\rho_{CO_2}$ difference of <0.01 (Fig. 6d). In estuaries, however, the salinity can vary from 0–35, which implies a decrease in $\rho_{CO_2}$ from 0.95 to 0.86.

The atmospheric $CO_2$ concentration also affects $\rho_{CO_2}$. When atmospheric $CO_2$ levels increase, more $CO_2$ moves into solution to maintain the equilibrium between water and air, as described by Le Chatelier's principle. An increase in the atmospheric $CO_2$ concentration of 100 ppm would increase $\rho_{CO_2}$ by ~0.004.

In conclusion, $\rho_{CO_2}$ is easily calculated from environmental parameters, and is maximized in systems with low temperature, pH, and salinity. However, the precise value for $\eta_{A_T}$ is difficult to quantify, but likely equates to ~0.8–1 in well mixed shallow systems. It is paramount that mERW is applied in areas where alkalinity is not lost to the deep ocean (He and Tyka, 2023).

## 4 Conclusions and future outlook

CDR technologies are urgently required to meet the targets of the Paris Climate Agreement, and OAE through mERW is

considered a promising candidate (Meysman and Montserrat, 2017). A considerable advantage of mERW is the possibility of rapid deployment and scalability, as no new technologies need to be developed. Since mERW mimics the natural process of chemical rock weathering, its theoretical underpinnings are well understood: when fine-grained silicates are added to coastal sediment environments, they remain out of thermodynamic equilibrium and will dissolve, thus releasing alkalinity (Fig. 2a). Although mining efforts would have to be substantially expanded to supply olivine for mERW, potential dunite reserves are

relatively large, so resource availability does not appear to be a limiting factor. Furthermore, the deposition of olivine onto the seafloor is expected to increase the dissolution rate of olivine by exposing it to the benthic weathering engine: chemical weathering can be enhanced by wave-induced physical abrasion, removal of weathering products through advection and bioirrigation, and exposure to a lower pH (compared to the overlying water). Although initial studies are positive (Flipkens et al., 2023b), the efficiency of this benthic weathering engine under relevant natural conditions remains poorly quantified, and

so this is an important point of attention for future mERW studies.

Coastal environments are geochemically complex and heterogeneous, so the practical implementation of mERW requires consideration of many processes, some of which are still poorly constrained (e.g. the impact of saturation effects in the pore water in different types of coastal sediments). Moreover, the intrinsic geochemical difference between various types of coastal sediments (e.g. cohesive versus permeable versus bedload) has been given very little attention. In addition, much of our current

understanding of the efficiency of mERW is based on laboratory experiments in aqueous solutions water with high fluid-to-mineral ratios, and it is unknown to what extent these results can be extrapolated to natural mERW field sites. Here, we have

provided a systematic review of the parameters that determine the $CO_2$ sequestration rate of olivine-based mERW (Table 4) and have identified aspects that need consideration when applying ERW in coastal environments.

**Table 4. Summary of predictability of each parameter needed to calculate the $CO_2$ sequestration rate from mERW (basis for classification in supplementary file 1.4).**

| Parameter | Symbol | Predictability |
|---|---|---|
| **$CO_2$ sequestration efficiency** | $\rho_{CO_2}$ | High |
| **Molar mass olivine** | $M_{olivine}$ | High |
| **Mass fraction of inert minerals** | $x_{inert}$ | High |
| **Alkalinity transfer efficiency** | $\eta_{A_T}$ | Moderate |
| **Intrinsic olivine dissolution rate** | $k_d$ | Moderate |
| **Specific surface area** | $A_{surf}$ | Moderate |
| **Alkalinity production factor** | $\gamma_{A_T}$ | Low |
| **Saturation factor** | $\Omega$ | Low |

Table 4 lists the key parameters that affect the dissolution of olivine and the associated $CO_2$ sequestration efficiency and provides an assessment of their current predictability (i.e. how well we can predict them for a given environment with current knowledge). The $CO_2$ sequestration efficiency $\left(\rho_{CO_2}\right)$, mass fraction of inert minerals $\left(x_{inert}\right)$, and molar mass of olivine $\left(M_{olivine}\right)$ are classified as "highly predictable" as they can either be calculated from environmental data ($\rho_{CO_2}$) or can be accurately derived by chemical analysis of the source rock of olivine ($x_{inert}$ and $M_{olivine}$). The alkalinity transfer efficiency $\left(\eta_{A_T}\right)$ can be estimated using coupled chemical-hydrodynamical models (e.g., Daewel and Schrum, 2013; He and Tyka, 2023), if such models are available at sufficient resolution for the application site. Furthermore, the prediction of the dissolution rate of olivine in actual coastal sediments is hampered by several uncertainties, and so the predictability of the associated parameters; the intrinsic olivine dissolution rate $\left(k_d\right)$, and specific surface area $\left(A_{surf}\right)$ are qualified as "moderate". The intrinsic olivine dissolution rate is well-studied in laboratory settings, but its actual value within actual marine sediments bears considerable uncertainty. Experiments with a particular focus on mERW are typically conducted in laboratory reactors with high fluid-to-sediment ratios. Little information on the intrinsic olivine dissolution rate within actual sediments is available, which makes it difficult to estimate  for the three suggested mERW scenarios (bedload, permeable, and cohesive sediment applications). Therefore, attaining estimates of the intrinsic dissolution rate in different natural sediment settings should be a research priority. As pH is a critical parameter in determining the intrinsic dissolution rate, an improved understanding of pore water pH as a function of sediment type would be beneficial in assessing which application sites are promising for mERW. Furthermore, a more detailed and systematic monitoring of the specific surface area of the olivine grains seems adamant. The specific surface area and grain roughness are typically only measured or reported before the start of dissolution experiments;

measurements throughout (long-term) applications could give important information on how the reactive surface area changes during mERW application.

Another important parameter governing olivine dissolution is the saturation factor $(\Omega)$, for which the predictability is qualified as "low" due to the low amount of data currently available. The uncertainty relates to the residence time of the pore water, which is determined by an intricate interplay between the grain size and properties of the ambient sediment, the local hydrodynamics, and the benthic fauna community composition. It seems imperative to mERW that olivine is applied to well-flushed sediments so that alkalinity and weathering products do not accumulate (hence $\Omega \ll 1$).

Also, the predictability of the alkalinity production factor $(\gamma_{A_T})$ is qualified as "low". The formation of the clay minerals sepiolite and serpentine seems unlikely in coastal sediment conditions, however, the formation of clay mineral assemblages (iddingsite and smectites) has not been thoroughly addressed. The prevalence of carbonate precipitation reactions is uncertain. Initial studies suggest that carbonate precipitation would be limited and that the risk of runaway precipitation (in the sediment or water) during mERW is low since alkalinity is released slowly and is added to an extensive volume of water. However, a critical unknown is how mERW would impact natural carbonate dissolution within the sediment and whether the alkalinity generated via mERW is fully additive to the natural alkalinity generation (Bach, 2024).

Considering the factors in Table 4, the suitability of certain sites for application of mERW can be evaluated. The most ideal locations are areas that are warm or relatively acidic, as these factors contribute positively to the weathering rate of olivine through the intrinsic dissolution rate. For mERW, it is key that sediments are targeted where olivine dissolution products do not build up, to avoid secondary mineral formation or inhibition of natural alkalinity production. However, the balance between removal of dissolution products and sufficient build-up of acidity to enhance the olivine dissolution should be considered with regards to the water exchange rate. Additionally, it is important that the generated alkalinity is able to reach the surface water and equilibrate fully with the atmosphere, implying that coastal waters should be well-mixed and that olivine is not deployed in areas characterized by downwelling to the deep sea. Furthermore, olivine mines should be close to the coast to limit costly overland transport and avoid transport emissions (which negatively impacts the $CO_2$ sequestration efficiency).

Overall, we conclude that whilst studies show a clear potential for mERW as a CDR method, the approach is not ready for upscaling and commercial exploitation. Future experiments should focus on improved quantification of olivine dissolution rates as well as alkalinity release rates in actual sediment environments. There is hence a clear need for *in situ* field trials, which are essential to quantify how efficiently mERW performs as an OAE technology. Before field trials are launched, microcosm and mesocosm experiments with natural sediment are crucial to increase the predictability of mERW effects. Both alkalinity effluxes from the sediment as well as the chemical conditions in the pore water before and after the application of olivine should be monitored. Since the effect of mERW is expected to vary seasonally, measurements should have a sufficiently high temporal resolution to allow trends to be quantified for MRV. Olivine grains should further be recovered throughout experiments to verify olivine dissolution features, formation of passivating layers, and occurrence of secondary mineral reactions. Detailed evaluations of the ecotoxicological and environmental impacts of mERW are another prerequisite for field

trials. Accordingly, a critical next step for mERW is to determine the efficacy of $CO_2$ drawdown in real-life conditions and its impacts on the broader environment. Moreover, the outcome of these field experiments will determine MRV protocols, which need to be designed and agreed upon, so that mERW applications can proceed in a sustainable and economically feasible manner.

**Author contribution**

LG was responsible for formal analysis, data collection, visualization, and writing of the original draft. LG and FJRM were responsible for the conceptualization. FJRM conceived and supervised the study. LG and FJRM took the lead in writing with additional input from AH. All authors discussed the results and contributed to the final manuscript.

**Competing interests**

The authors declare that they have no conflict of interest.

**Financial support**

LG was supported by a PhD fellowship from the Research Foundation – Flanders (FWO) (project number 1S08321N). AH was supported by a junior postdoctoral fellowship from FWO (project nr 1241724N). FJRM was supported by the Strategic Basic Research (SBO) project from FWO (project nr S000619N). This research was financially supported by Team for the Planet (https://team-planet.com), a citizen community initiative dedicated to tackling the climate challenge (funding through a collaboration agreement with its Carbon Time subsidiary).

**Acknowledgments**

The authors thank Sebastiaan van de Velde and Gunter Flipkens for academic discussions, and Vincent Sluydts for his assistance with calculating the uncertainty for the intrinsic dissolution rate constant. Ivan Communod and Mathieu Helwig from Carbon Time are thanked for their support through Team for the Planet.

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
