# Peer review of "Review and syntheses: Ocean alkalinity enhancement and carbon dioxide removal through marine enhanced rock weathering using olivine"

_EGUsphere, 2024_

## Author Response (AR1)

*We are grateful for the constructive and positive feedback on the manuscript provided by the reviewers. Their recommendations have significantly improved the structure and content of the text. We have carefully considered and addressed all comments; our responses are in italic. Line number changes refer to the lines found within the "Author's track changes file". We have supplemented to our previous response to the reviewers the specific line changes where the changes can be found. Our original response to the reviewers' comments have not changed, besides line references to now correctly reference the "Author's track changes file".*

**1    Reviewer 1**

The manuscript by Geerts et al. provides an overview of different types of coastal applications using enhanced silicate weathering as carbon dioxide removal method. Different coastal areas (ranging from bedload movement to permeable and cohesive sediments) are discussed with their respective advantages and disadvantages. The impacts of relevant geochemical parameters, such as pH and temperature, are modelled for the distinct scenarios and the uncertainties as well as knowledge gaps emphasized. While the manuscript provides a nice overview of the current state of knowledge, I'm missing an outlook for the most promising areas of application despite the current knowledge gaps. I would appreciate an effort to model the deposition of olivine in the different coastal environments taking also into account the accessibility of the deployment site and feasibility in the sense of political will and economics into account. This modelling effort could also take into account the different temperatures in the vicinity of the equator versus the lower solubility in comparison to northern and southern latitudes.

*We agree that an in depth exploration of suitable deployment sites, a full techno-economic analysis as well as a discussion societal, legal and political considerations, would be highly interesting and worthwhile. However, this would require a very large endeavour, which goes way beyond the normal revision. Moreover, it would imply the collection and presentation of a large mount of information on top of what's already presented now. Given that the present manuscript is long, we consider this is outside the scope of the present review. Here, we'd like to focus on the geochemical feasibility of CDR via marine enhanced rock weathering (mERW) with olivine, which will e.g. provide crucial knowledge for a downstream techno-economic analysis.*

*Still, we have taken the remark as a crucial point of attention in the revision, and we now stress - where possible and relevant - the importance of geographical location throughout the manuscript (e.g., section 2.2 "Availability of olivine", section 4 "Conclusions and future outlook").*

*Changes: Line 324-328, added discussion on the effect of olivine source location for mERW. Line 1040-1042, we have added that for the parameters, $p_{CO2}$ and $\eta_{AT}$, mERW is most effective in low temperature, salinity, and pH systems. Additionally that mERW should be applied in places where no alkalinity is lost to the deep ocean.*
*Line 1116-1125: considering the factors discussed throughout the conclusion, we added a small discussion on what the best general locations would be for mERW.*

Next to this more elaborated outlook, I am missing some literature or would replace literature investigating enhanced rock weathering in the terrestrial environment with publications explicitly investigating the minerals of interest in the marine environment (see comments below). As this

manuscript is a review, I would carefully check all references and verify their applicability, especially in the context of marine coastal OAE.

Nevertheless, I enjoyed reading the manuscript and recommend publication after revising the two major points mentioned above and minor comments below.

*We have carefully checked the manuscript, and where needed, we have changed references to studies from marine environments. In some cases, references to terrestrial ERW have remained due to a lack of studies in marine systems. In such cases, we now explicitly mention that these studies dealt with terrestrial ERW.*

*Changes: Lines 54, changes reference of Meysman and Montserrat (2017) to Archer et al. (2009).*
*Line 66, added references to Eisaman et al., (2018) and Rau et al., (2018).*
*Lines 86, 121, 235, 505, 655, 659, 946, and 951, added reference to Fuhr et al., (2024).*
*Line 83, removed reference to te Pas as it was a terrestrial study, added references to (Bach et al., 2019, Hartmann et al., 2013, Renforth and Henderson, 2017) instead.*
*Lines 116 and 173, removed reference to Schuiling and Krijgsman (2006)*
*Lines 224, 603, 935 added reference to Li et al., (2024)*
*Line 630 reference to Meysman and Montserrat (2018) was removed as it was not an experimental study.*

*Section 1.2 was restructured, from context it is now clear when we are discussing these minerals in a general sense, specifically in the context of mERW or in the context of terrestrial ERW.*

Line 23: Typo field conditions

*This has been corrected.*

*Changes: Line 24*

Line 45: With respect to the cited literature and to my best knowledge, the estimated lower potential is 0.5Gt CO2 yr-1 instead of 0.1Gt CO2 yr-1.

*The reviewer is right that there are many different $CO_2$ sequestration potential ranges reported in literature. The range used in the manuscript is derived from NASEM (2022) which reported a uptake potential of >0.1 to 1 Gt $CO_2$ $yr^{-1}$. We have added a ">" that was previously omitted from the text.*

*Changes: Line 53*

Line 48: I assume, Fig. 1a is not the correct reference here. These olivine beaches constitute more the exception on Earth's surface than the norm as you also show in your map in Fig. 3. Either refer to Fig. 1b or put a picture of a mountain drained by rivers.

*Figure 1a has been removed following comments from both reviewers.*

*Changes: Line 93 Fig. 1*

Line 50: What is meant by organic C sequestration? Blue carbon or enhanced primary production through fertilization? Please add a short explanation.

*Organic carbon sequestration encompasses both blue carbon and enhanced primary production, as well as land-based approaches such as afforestation. We omitted the mention of organic carbon sequestration to avoid confusion, as pH buffering is a benefit unique for OAE. We have clarified this:*

*"Moreover, OAE has the important benefit of counteracting ocean acidification, which is not the case for other CDR techniques that only target $CO_2$ sinking, such as reforestation on land, or blue carbon and ocean fertilization in the marine environment (Campbell et al., 2022; Caserini et al., 2022; Meysman and Montserrat, 2017)."*

*Changes: Lines 58-60*

Line 53-56: The publications by Eisaman et al. (2023) and/or Rau et al. (2018) should also be cited here?

*Thanks for the suggestion. We have added both references.*

*Changes: Line 66*

Line 72: The publication by te Pas et al. (2023) investigates ERW in the terrestrial environment and conducted experiments solely in the context of soil applications? I highly doubt that mineral dissolution rates are comparable in soils and seawater matrix and I definitely wouldn't cite this publication here (and throughout the manuscript), in a study focusing explicitly on coastal marine applications. Also the other two cited papers don't appear to be very well fitting for OAE. I would rather cite publications explicitly investigating different mineral in the context of OAE, such as Hartman et al. (2013), Bach et al. (2019), and Renforth and Henderson (2017).

*We agree with the reviewer and have removed te Pas et al. (2023), as well as Huijgen et al. (2006), as their focus was on mineral carbonation. In their place we added references to Hartman et al. (2013), Bach et al. (2019), Renforth and Henderson (2017), and NASEM (2022).*

*Changes: Line 126, references changed to include Renforth and Henderson for wollastonite. Section was moved to 1.2 as part of the restructuring. The references were changed as they are more fitting, the section now focuses on silicates for enhanced rock weathering in general.*

Line 74-75: why did you choose to concentrate on CESW via olivine addition in your review and not on the other methods discussed before? Motivation is not clear.

*We focus on olivine as it is the mineral that has so far received most interest in the context of marine ERW. This interest stems from its fast weathering rate, high $CO_2$ uptake rate and relative abundance. We have further clarified this and stress the motivation to focus on olivine earlier in the introduction, and discuss the use of olivine relative to other minerals already in section 1.2.*

*Changes: Lines 85-89, and lines 123-143 now more clearly motivate the choice of using olivine comparing it to other minerals.*

Line 93: I wonder, if it is actually possible to compensate anthropogenic CO2 just by 'sitting it through', if emissions continue as business as usual. I would add a comment, that this scenario is only possible when emissions are cut drastically.

*We agree with the reviewer. When emissions are halted, an excess of $CO_2$ will still be in the atmosphere. This excess of $CO_2$ will eventually be sequestered by natural silicate weathering, but this is only possible when emissions are halted. This has been clarified:*

*"However, the timescale of this response is too slow (> 10,000 years) for society. Even if emissions were completely halted, we would have to "sit through" an extended period of global warming before the excess of anthropogenic $CO_2$ is removed naturally (Archer et al., 2009)."*

*Changes: Line 48-50*

Line 148: Also Fuhr et al. (2024) investigated OAE with sediments in their experiments.

*The study by Fuhr et al. (2024) has been added here and throughout the manuscript where relevant.*

*Changes: Lines 86, 121, 235, 505,655, 659, 946, 951 added reference to Fuhr et al., (2024).*

Line 189: Error in the anorthite formula and also here, I would add a comment that dissolution rates of these minerals were not investigated in seawater matrix and might be lower.

*The formula for anorthite has been corrected, and we now specifically mention in the text that these rates are measured in freshwater laboratory experiments. The text has been moved to section 1.2.*

*Changes: Line 126, corrected formula. Line 146, Table 1 now specifies that these rates have been measured in freshwater under well-mixed conditions and that in seawater and sediments, the dissolution rate may be slower.*

Line 238 ff: I doubt that this paragraph is very relevant, as the occurrence of olivine beaches is globally very low (as indicated in your Fig. 3). I don't think, anyone is considering this option seriously for CDR, so I would delete this paragraph and also the two points on the map in Fig. 3.

*We have removed this paragraph and edited Fig. 3 accordingly.*

*Changes: 330-334 was removed. Fig. 3 was changed to no longer include the beaches.*

Line 334: I would state more explicitly, that Rimstidt et al. defined a pH of 5.6 and higher as basic, as a pH below 7 is normally not described as basic.

*We have clarified this in the text: "The rate equation was solved for pH 3.13 and 8.22, respectively, which were the average pH values of experiments which Rimstidt et al. (2012) classified as "acidic" (pH < 5.6) and "basic" (pH > 5.6)."*

*Changes: Line 445-446 adjusted accordingly.*

Line 387 and 595: The experiments by Fuhr et al. (2023, 2024) were conducted at low T (~10°), alkaline pH, and in the 2024 publication also under anoxic conditions.

*The experiments by Fuhr et al. (2023, 2024) and Flipkens et al. (2023) were indeed conducted at low temperatures and seawater pH, but they did not report $E_a$ values which is the focus of line 387. However, we have included references to Fuhr et al. (2023, 2024) and Flipkens et al. (2023). The text now reads as: "Only recently have studies been conducted at environmentally relevant conditions (pH ~ 8 and temperature 0–25°C, e.g., Flipkens et al., 2023b; Fuhr et al., 2023, 2024). However, $E_a$ was not reported in these studies. The value of $E_a$ is thus still uncertain for natural conditions, pinpointing an area for further research."*

*We have revised line 944, now reading as: "Most mERW experiments have been conducted in oxygenated seawater and are not fully representative of sedimentary conditions (e.g. in terms of $O_2$ and pH), and studies in sediments under hypoxic-anoxic bottom water have not investigated the formation of secondary minerals (Fuhr et al., 2024)."*

*Changes: Lines 505-509 and lines 944-946*

**Line 479:** Would be nice to include a comment about cable bacteria here: Fuhr et al. 2023 with reference to Meysman et al. (2019). In addition, the recent study by Li et al. (2024, https://doi.org/10.1016/j.scitotenv.2023.168571) investigates the impact on diatoms by olivine dissolution as well as the effect that diatoms have removing the passivating layer. This study should definitely be included here.

*We have added a brief mention of the effect of cable bacteria through a pH change in section 3.1.2 (Impact of pH on olivine dissolution), as the focus of section 3.1.6 is specifically on passivating layer formation.*

*The study by Li et al. (2024) has been added and is discussed briefly in section 3.1.6.*

*Changes: Line 190, and lines 602-605*

**Line 580:** Why not turn it around then? Saponite (former bowlingite).

*We agree, this has been rewritten and clarified.*

*Changes: Line 754*

**Line 607 to 625 + 657:** I don't understand the separation of Section 3.3 from the previous sections, as it also deals with secondary precipitates. Also, the formation of secondary clays can play a major role during CESW, as described for example in Griffioen (2017) and Fuhr et al. (2023), so I wouldn't exclude it here. The additionality problem described by Bach (2024) is indeed important and should be highlighted. As you do this in line 657ff, I would recommend to delete this paragraph here to avoid repetition.

*The separation of sections 3.2 and 3.3 was made to distinguish between reduction of the net alkalinity release due to incomplete dissolution of olivine (less alkalinity is being produced) and precipitation of dissolution products (alkalinity produced through dissolution is consumed).*

*However, following recommendations from both reviewers, we have merged sections 3.2 and 3.3 as for some secondary minerals, it is not clear which pathway has led to their formation. In merging the sections, we have also restructured and clarified the discussion about clay formation.*

*The additionality problem is now only described in section 3.2.4 ("Stimulated precipitation and inhibited dissolution of carbonates").*

*Changes: Former sections 3.2 and 3.3 have been merged into 1 section "3.2 Secondary mineral formation". The formulas on line 361, 387 were edited to incorporate this change. Also line 256 Table 2 was changed. As well as, lines 1066 Table 4, and lines 1099-1105 were changed to reflect this change.*

**Table 4:** In this sequence, it is not clear why you use fayalite instead of forsterite here, as fayalite makes up 'only' 6-20mol%, as you mention in line 684? I would bring this argument (your lines 684 to 688) earlier or make a comment in the caption of Table 4.

*We have clarified the table caption: "The $Fe^{2+}$ stems from fayalite, as olivine typically comprises of 6–20 mol% Fe.*

*Changes: Line 681*

**Line 781ff:** Personally, I'm not a big fan of referring to symbols instead of descriptions in flowing text. The repetitive look up for explanations disturbs the reading flow.

*We have added the full parameter names.*

*Changes: Lines 1069-1105 were edited accordingly.*

**Figures**

Fig. 1: (a) As nice as this picture is, I don't think it adds much value to the discussion and I would remove the discussion about olivine beaches completely from this manuscript. (b) what about alkalinity-consuming process, e.g. carbonate formation?

*Following the suggestion from the reviewers, we have removed Fig. 1a. Fig. 1b was removed according to a suggestion from reviewer 2.*

*Changes: Line 92 Fig. 1*

**Fig. 2a:** I would include phyllosilicates here as well.

*Phyllosilicates formation has been added to the figure.*

*Changes: Line 203, Fig. 2a*

**Figure 3:** Would be nice to include in this map, where currently dunite is mined and which of the deposits are realistic candidates for future exploration considering social, politic, and economic aspects.

*We agree with the reviewer that information about operational mines would be a relevant addition to the figure. However, while it is well known where certain commercial mines are operating (e.g., Norway, Spain, Italy, Turkey, Japan), information about the exact reserves that are exploited at the moment is not always available. To avoid confusion, we have opted not to include this feature in the figure. A discussion about social, political and economic considerations of dunite mining is beyond the scope of this review.*

*Changes: Not applicable*

**Fig. 4a:** is there a reason, why saline data points in the legend are much larger compared to the other? What are 75 Cl and 95Cl? Explanation missing in caption.

*Saline points are enlarged as these data are most relevant for marine ERW, emphasizing that dissolution rate data at seawater pH are lacking. The explanation of "CI" had indeed been omitted in the caption, we have now clarified that it stands for confidence interval.*

*Changes: Lines 443-444 Fig. 4.*

**2     Reviewer 2**

The manuscript provides a review of coastal enhanced weathering with olivine minerals, and investigates the state of the art on many of the key factors which determine the rate of olivine dissolution, production of alkalinity, and conversion of alkalinity into actual CDR. I appreciate the effort but think the manuscript would be improved by a more complete overview of coastal enhanced weathering before focusing in on olivine. In addition, I think there are a number of important missing discussion points from the review. I also believe the manuscript would be improved with some restructuring. I elaborate on these points in detail below.

L8 - The authors use the term "coastal enhanced silicate weathering" (CESW) throughout the text. At this point, I think the field has essentially landed on "enhanced rock weathering" (ERW) as the name for this CDR strategy. If the authors wanted to distinguish coastal ERW specifically, I would therefore abbreviate it as CERW. I also would not reference "silicate" weathering specificalyl, given that CERW with non-silicate minerals is under consideration as well. Its unnecessarily limiting. And anyway "CESW with olivine" is redundant since olivine is a silicate. Altogether, my recommendation is "CERW with olivine" to be consistent with the field.

*We have followed the suggestion of the Reviewer. We now use the term "marine ERW" rather than "CESW". We have followed the suggestion of the Reviewer. We now use the term "marine ERW" rather than "CESW".*

*Changes: Changed "CESW" to "mERW" throughout the text.*

**L46** - Rather than citing papers that state storage timescales by referencing yet other papers, I recommend citing the actual papers that establish such estimates of timescale, e.g. David Archers work, Jack Middelburgs work, etc.

*We have replaced Meysman and Montserrat (2017) with Archer et al. (2009).*

*Changes: Line 54, reference was changed.*

**Figure 1** - I would cut images a) and b) from this figure. With regards to a), the image of the beach in Hawai'i and natural olivine beaches aren't discussed in the text and so no context is given. Plus you cant see any olivine in this picture anyway, so what is it adding? With regards to b) the natural weathering process has been covered at length elsewhere. For a review like this, I would simply assert that CERW is based on the natural process, provide references and move on. Its not a meaningful contribution to this article.

*Following the suggestion from the reviewers, we have removed Figs.1a and 1b. Additionally, we have restructured the introduction and decreased the focus on natural weathering.*

*Changes: Line 93, Fig. 1a and b were removed.*
*Line 102-110 was removed to reduce focus on natural weathering.*

**L88-89** - "The chemical dissolution of silicate minerals in terrestrial and coastal environments releases alkalinity into the surface ocean, where it drives the uptake of $CO_2$ from the atmosphere" This is an incorrect summary of how ERW works on land. On land, the alkaline minerals react with, and sequester atmospheric $CO_2$ directly and the ocean is simply a holding tank for the resulting alkalinity. This is different from ERW in the ocean, where the alkaline minerals react with, and sequester seawater $CO_2$, which ultimately then drives $CO_2$ from the atmosphere through air-sea gas exchange.

*We agree with the reviewer. After rewriting and shortening the text about natural silicate weathering, the line has been removed from the text.*

*Changes: Lines 103-110 were removed.*

**L100** - Remove Schuiling and Krijgsman, 2006 reference. This paper did not suggest the application of silicates to coastal and shelf environments, only land.

*We have removed the reference.*

*Changes: Line 116, removed reference.*

**L100 - 119** - I take issue with the framing the authors present here that there are three distinct CERW scenarios. It does not appear to be based on an understanding of sediment transport and coastal geomorphology. To begin, for the "bedload scenario" there is essentially no real-world scenario where a coastal environment is made of only gravel. Coastlines exist with a gradient of particle sizes. There will always be some fine grained material, and any olivine sand that was added to a rocky coastline with almost immediately sort, based on the brazil nut effect, downwards to those finer native particles. Regarding the permeable and cohesive sediment

scenarios, I would argue that the permeability (and porosity (Köhler et al., 2013)) of a sediment are simply factors which should be discussed as controls on advection, bioirrigation, as well as diffusion (the latter which was excluded altogether from this manuscript). Coastal sediment will have some component of all three of these processes occurring. Permeability and porosity influence how the relative importance, and magnitude, of each of these three processes differ across coastlines. And of course, other factors will influence advection, bioirrigation, as well as diffusion as well, such as local oceanographic conditions and organic matter content. Taken together, I think the presented framing is overly simplistic and somewhat inaccurate (e.g. diffusion), and the authors would be better served by instead including a discussion of the impact of permeability further on in the paper when they discuss other controls on dissolution rates.

*The abstraction into three sediment environments provided a simplified description of the reality of coastal systems, but a useful one. The aim of this manuscript is to discuss factors that affect olivine dissolution in coastal environments. To this end, we provide a framework with three main types of sediment environments found in coastal systems. We do not contend that a single coastal system is solely made up of one of these types. For example, gravel beds do occur in the North Sea, along side more permeable sandy areas, and localized areas with deposition of cohesive muds (Wadden Sea, German Bight). The abstraction into three sediment types is valuable for illustrating the dominant controls on marine ERW exerted, as each sediment type has its own specific transport regime.*

*However, we agree with the reviewer that the rationale for doing this abstraction could be substantially improved. We now start the discussion by describing how hydrodynamic energy regulates the environmental conditions (grain size, permeability, organic matter, pH) and main transport process (advection, also bioirrigation, vs diffusion) in sedimentary systems, as these factors are strongly correlated with each other. For this reason, we believe that for the purposes of marine ERW, coastal systems globally can be categorized into three broad categories.*

*Changes: Lines 150-169, a new section was added (section 1.3) which first discusses the rationale for this abstraction.*

Additionally, this framing leaves out entirely the role for water column dissolution in CERW. Any CERW that uses sufficiently small particle sizes will have some fraction of the olivine dissolution occur in the water column through particle resuspension (i.e. suspended sediment). Furthermore, some scientists are working on understanding CERW project sites to intentionally increase the amount of water column dissolution so as to avoid the messy complications of sediment dynamics. To date, much of the CERW modeling has actually assumed water column dissolution for small particle sizes (e.g. Feng et al. 2017). Water column dissolution - and what, for example, water column pH means for dissolution - should be added to this review.

*Water column dissolution is captured by the bedload scenario, where dissolution occurs in ambient seawater (and not porewater with composition that is distinct from the overlying water column).*

*The reviewer is correct that modelling studies have focused on olivine weathering in the water column. Yet, we are not aware of any publications about experimental studies on olivine based ERW that aim to intentionally increase the amount of water column dissolution.*

*However, there are many problems and challenges with "intentional" water column dissolution (1) the energy required to produce sufficiently small grain sizes and keep olivine in suspension is very large, which reduces the $CO_2$ drawdown efficiency considerably (2) fine-grained particles will be also be transported and deposited into cohesive sediment sites.*

*We address this in section 1.2, as a motivation for why most research now target sediment-based ERW approaches.*

*Changes: Lines 116-121, comment on the required grain size for water column mERW, and motivation for sediment based mERW.*

Finally, this discussion also leaves out that many coastal environments under consideration for CERW are not beaches or similar sandy coastlines, but rather wetlands - marshes, mangroves, etc. There is a funded field trial of this in the US right now. This should be included in the discussion. https://oceanacidification.noaa.gov/funded-projects/tidal-wetlands-as-a-low-ph-environment-for-accelerated-and-scalable-olivine-dissolution/

*In this framework, wetlands, saltmarshes and mangroves would be categorized as cohesive sediments.*

*Changes: Line 160-167 discusses the processes at play in these systems indirectly (under "cohesive sediments").*

**Figure 2a** - should include a depiction of clays as secondary minerals as well, beyond carbonates and metal oxides.

*Phyllosilicates formation has been added to the figure.*

*Changes: Line 203 Fig 2a, phyllosilicate formation has been added to the figure.*

**L135 - 140** - I would reference here the extensive amount of work on ecology and ecotox that has been done, but simply state that it is outside the scope of the review.

*We have added references to the ecotoxicology work that has been done. The sentence now ends as: "..., olivine dissolution rates (Hangx and Spiers, 2009; Heřmanská et al., 2022; Oelkers et al., 2018; Rimstidt et al., 2012), and ecotoxicology and ecology impacts (Bach et al., 2019; Flipkens et al., 2021, 2023a; Li et al., 2024)."*

*Changes: Lines 223-224.*

**L142** - You reference here "Vesta, 2023" which is a monitoring report, but in the next sentence you say there are no reports on the outcomes of field trials. It seems like there is a report on outcomes, since you reference it. Additionally, Vesta (and their collaborators) have generated a lot of conference abstracts, which I recommend you reference as well. The USGS also did a field trial of olivine in a coastal environment (see link below), the PI on this field trial is Kevin Kroeger. His team has also published a lot of abstracts on their findings at AGU and OSM. Both this field trial and key abstracts should be mentioned. I would also reference the field trial conducted by Planetary Technologies. They added the mineral brucite to the seafloor in Halifax

Harbor, Canada. Its not olivine, but it is nonetheless a CERW field trial that warrants mention given how few are being conducted. Theyve published abstracts as well with Dalhousie University. In general, given that coastal enhanced weathering field trials are ongoing, I am surprised how little attention they receive in this manuscript, particularly given that "field trials are needed" is a primary conclusion.

https://www.usgs.gov/media/images/usgs-and-partners-collect-a-soil-core-massachusetts

*The reviewer is correct that several field trials are ongoing, we have added this information in section 1.3. However, no peer-reviewed publications have emerged from these trials. We have deliberately chosen to only cite results from peer-reviewed studies and do not include conference abstracts.*

*Changes: Lines 225-227, clarified and added references to the projects suggested by the reviewer.*

**L145** - You said these laboratory experiments are done under "idealized conditions". I would argue that these are definitely not "ideal" given that they are closed systems and subject to batch effects, as discussed later in the manuscript.

*We have removed the mention of "idealized" laboratory conditions throughout the text.*

*Changes: Lines 15, 231, 644, and 1060, changed accordingly.*

**L178** - Also cobalt.

*We have clarified that nickel and chromium are not the only trace metals in olivine, but the two main ones. The sentence now reads as: "Olivine also contains trace metals, notably, nickel (Ni) and chromium (Cr). Ni substitutes the divalent cations in olivine ($Mg2+$, $Fe2+$), and its content ranges from 0.2–1.2 mol% (Keefner et al., 2011; Montserrat et al., 2017; Santos et al., 2015).". Although olivine contains some cobalt, the concentrations are very low (0.01 mol%; Herzberg et al., 2016), so we do not mention this metal specifically.*

*Changes: Line 264-266, clarified.*

**L190 - 195** - Since you have expanded the scope beyond just silicates here (e.g. carbonates) I would also mention brucite (particularly since Planetary already did a field trial with this mineral, as mentioned above), and anthropogenic minerals too. I think that would produce a better synthesis of the CERW field generally.

*As part of the restructuring of the introduction we have moved the comparison of olivine to other rocks to section 1.2, where we also discuss brucite.*

*Changes: Lines 274-283 were removed and rewritten as part of restructuring. Now the comparison happens in section 1.2, on the lines 123-142.*

**Figure 3** - On this figure you have a large olivine deposit labeled in Colorado. Is this supposed to be the Twin Sisters deposit? If so, the Twin Sisters deposit is in Washington State, not Colorado.

*The reviewer is correct, we have edited the map.*

*Changes: Line 329, Fig 3c, edited the figure accordingly.*

**L251 - 253** - Transport (shipping) of the olivine to the project site is also an important step between production of olivine (mining and grinding) and spreading the olivine. Please include.

*We agree with the reviewer and have included the transport step. The section now reads as:*

*"The entire process of marine ERW can be quantitatively described as a six-step process (Fig. 2a): (1) production of olivine sand from olivine rich source rock (2) transport of olivine to the project site, (3) spreading and deposition onto the seabed (3) mineral dissolution, (4) alkalinity release to the overlying water, and (5) $CO_2$ sequestration at the air-sea interface. In the dissolution step, the olivine sand deposited during marine ERW will react on top of or within the seabed."*

*Changes: Line 343 added the transport step.*

**L257** - The equation here presents Rdiss as if there is a single dissolution rate attached to the feedstock applied to the seafloor, but further down (L268) the text acknowledges that any feedstock has accessory minerals that don't contribute alkalinity, though the text should include the possibility that the accessory minerals simply contribute alkalinity with a ratio g/g ratio and a different rate. In other words, the feedstock (not a singular mineral) applied to the seafloor will be a mixture of different minerals, with different CDR efficiencies and different dissolution rates but this is not made explicit.

*$R_{diss}$ is defined as the specific mineral dissolution rate, meaning that when a rock containing many different alkalinity-producing minerals (e.g. basalt) is utilized, a specific $R_{diss}$ is needed for each mineral present within the rock. We clarify this in the text: "When feedstock is utilized which contains an assemblage of different alkalinity producing minerals (e.g. basalt), each of these minerals will be defined by their mineral specific $R_{diss}$. An overall $R_{diss}$ for the entire feedstock can be calculated by averaging the mineral-specific $R_{diss}$ based on mass ratio."*

*Changes: Lines 355-357, were added.*

**L274** - Should also include a function that accounts for the potential of alkalinity loss in the water column to biotic carbonate formation, for example. This framework also leaves out the potential for incomplete air-sea gas exchange and a key factor that can reduce FCO2.

*We agree with the reviewer that incomplete air-sea gas exchange should be taken into account in equation 3 to 4, and have included a factor "$\eta_{AT}$" which stands for the alkalinity transfer efficiency. This is a time-dependent factor that denotes the proportion of bottom-water alkalinity available for air-sea gas exchange, and is further discussed in section 3.3.*

*Loss of alkalinity from marine ERW due to biotic carbonate formation in the water column is possible, but unlikely, and we now discuss this more in detail in section 3.2.3. Past marine OAE studies have not shown increase in biotic carbonate production (Ferderer et al., 2022), at the same time olivine addition promotes the growth of all phytoplankton and silicifying algae in particular, which outcompete carbonate-forming phytoplankton (Bach et al., 2019; Li et al., 2024), therefore, we do not include this as a factor in the equation.*

*Changes: Line 375 the formula was changed to include the new factor $\eta_{AT}$. Lines 978-989, and lines 1040-1042 now include a discussion on $\eta_{AT}$. Accordingly, lines 1066 Table 2 and lines 1073-1075 in the conclusion now include the factor $\eta_{AT}$. Lines 933-938 now includes a small discussion on biogenic $CaCO_3$ precipitation from the water column.*

**L284** - In general, this section and the overall manuscript should include a discussion of porewater CO2 as a limiting factor for olivine dissolution. If porewater exchange is low, or respiration is low (low organic matter), etc CO2 consumption may be the rate limiting factor for olivine dissolution.

*Past olivine dissolution experiments have demonstrated that the $CO_2$ concentration does not affect olivine dissolution rates directly, but only indirectly through changing the pH (section 3.1.1). The effect of pH on olivine dissolution, and drivers behind the pH (including the amount of oxic mineralization of organic matter), are discussed specifically in section 3.1.2 and throughout the manuscript.*

*Changes: Lines 476-480, further clarified the role microbes have on porewater pH specifically mentioning cable bacteria.*

**L298 - Section 3.1.1** - Generally speaking, I think this section would be improved by a discussion of the reasons why there is so much spread in the dissolution rate data (and therefore, how we can improve on it). For example, mineral impurities in the tested materials or the method of rate quantification (e.g. the use of Si concentrations which gives net dissolution rate) https://www.sciencedirect.com/science/article/abs/pii/S0016703719307811)

*We agree with the reviewer that this is an important point, and have modified the end of section 3.1.1: "The 95% confidence interval (CI) around the value of $k_d$ is large, spanning an order of magnitude log units at pH < 5.6 and several orders of magnitude at pH > 5.6 (Fig. 4a). This large spread in the data has previously been discussed by Oelkers et al. (2018) and Rimstidt et al. (2012), who attributed it to mineral purity (pure olivine dissolves faster e.g. Golubev et al., 2005), initial incongruent dissolution of olivine, lack of common data format and inconsistent reporting, differences in sample grinding and preparation, and most importantly, to inaccuracies in reactive surface area measurements section 3.1.5)."*

*Additionally, in section 3.1.5, we clarify that the inconsistency between BET surface area and mineral dissolution rates is one of the main reason for uncertainties: "Interestingly, this large increase in BET was not accompanied by a concurrent increase in dissolution rate (Grandstaff, 1978), highlighting that there can be an increase in the total surface area while the reactive surface area stays the same. Rimstidt et al., (2012) identified this inconsistency as the major reason for the large data spread found between dissolution experiments (Fig. 4a)."*

*Changes: Lines 434-438, discuss the reasons for $k_d$ uncertainty. Lines 581-584, discuss the inconsistencies between BET and mineral dissolution rate.*

**L506** - In general, its not clear to me why carbonate precipitation isnt included in this section on secondary mineral precipitation. Why distinguish between clays + metal oxides and carbonates? **L513** - "Clay formation has been proposed as a third potential mechanism…" but sepiolite is a clay. Youve just mentioned clay formation in the prior sentences. **L579** - This section is titled "formation of other clay minerals" but then talks about metal oxides. Though metal oxides are also discussed in section 3.3.2. Combine. **L607** - This "secondary reactions" section (3.3) is a combination of mineral formation (though excluding clays, which is section 3.2) plus a discussion of carbonate dissolution and additionality. My recommendation is to combine section 3.2 and the mineral formation part of section 3.3 into one section on secondary mineral formation, and then make section 3.3 strictly about mineral dissolution and additionality. I think this would make for a better logic flow.

*The separation of sections 3.2 and 3.3 was made to distinguish between reduction of the net alkalinity release due to incomplete dissolution of olivine (less alkalinity is being produced) and precipitation of dissolution products (alkalinity produced through dissolution is consumed). However, following recommendations from both reviewers, we have merged sections 3.2 and 3.3 as for some secondary minerals, it is not clear which pathway has led to their formation. In merging the sections, we have also restructured and clarified the discussion about clay formation.*

*Changes: Line 637 and onwards, merged prior sections 3.2 and 3.3 into one. Clarified the ways how alkalinity production can be decreased on lines 640-642. The tables and equations have been adjusted as well to reflect the new definition for $\gamma_{AT}$.*

**L658** - I think this discussion of carbonate dissolution would benefit from a brief review of where carbonate dissolution and the additionality question are likely to be significant (or not), as discussed in Bach, 2024. I think in the present manuscript, important context is missing that this additionality consideration is not likely to be important everywhere.

*We have expanded section 3.2.3 to include a description of the environmental conditions that may lead to additionality issues in marine ERW (sediments where oxic mineralization of organic matter drives a lowering of the pH) and discuss the impact on choice of location for olivine application.*

*Changes: Lines 903-909, added commentary for which types of sediments, $CaCO_3$ precipitation and the additionality problem may be an issue.*

**L672 - 682** - This paragraph, and discussion of water column processes, comes a bit out of nowhere. I would recommend that consideration of the water column should represent a unique step that must be assessed for CERW efficiency. This paragraph ends with the assertion that "The slow alkalinity release from the seabed during CESW implies a substantial dilution in the overlying water, thus avoiding local alkalinity excursions, and hence the chances of immediate CaCO3 precipitation in the overlying water are minimal." but this assertion isnt based on anything quantative. This would be substantially improved with even a back-of-the-envelope

calculation. I'll also add that referencing biotic calcification (which is not said specifically here) would benefit the manuscript.

*We now discuss biogenic $CaCO_3$ formation in the water column and runaway precipitation within the same paragraph, and have added a back-of-the-envelope calculation of the alkalinity release and resulting changes to $CaCO_3$ saturation state.*

*Changes: Lines 925-938, added commentary on both runaway precipitation and biogenic $CaCO_3$ precipitation.*

**L704** - This section should include a discussion of how water mass sinking and incomplete air-sea gas exchange also play a major role in determining CO2 drawdown - references such as He and Tyka, 2022 or Ho et al., 2023.

*We agree. The effect of water mass sinking and incomplete air-sea gas exchange on the $CO_2$ sequestration efficiency are now discussed in section 3.3.*

*Changes: Lines 982-989 and lines 1040-1042, we have included a new factor $\eta_{AT}$ that describes this incomplete air-sea gas exchange.*

---

## Author Response (AR3)

**Reviewer 1:**

Minor comments (line numbers refer to track changed manuscript):

*We thank the reviewers for their insightful comments, all line references refer to the new track-changed document.*

Figure 2a: any reason, why you leave iron sulfide precipitation out since you discuss it in detail in the secondary precipitation section?

*We only included the most important secondary reactions in the figure for visual clarity. mERW aims to utilize sediments that are sufficiently aerated to prevent saturation effects (which prevents FeS formation and so FeS formation is not included in the overview of Figure 2).*

Line 646: I would add 'pore waters' to the sediment, as this is where the dissolution products build up. '…contribute to the build-up of dissolution products in the sediment pore waters and promote…'.

*Changed accordingly.*

Line 656-659: The last part of this sentence is not correct. All of the cited publications on olivine dissolution in sediments refer to secondary precipitates, especially carbonate precipitation (Bach (2024) and Fuhr et al. (2024) observed no secondary carbonate formation and discussed the reasons; Fuhr et al. (2023) suspected it at the end of the experiment).

*The reviewer is correct that Bach (2024) and Fuhr et al. (2024) indeed suggested that secondary mineral formation may have occurred in their experiments, especially calcium carbonate formation, as the saturation index for calcium carbonate was elevated. However, our point was that calcium carbonate precipitation was not measured/demonstrated directly in these experiments (but indirectly inferred from ore water saturation state). We clarify on line 577-578.*

*"… and studies conducted on olivine weathering in sediment have indicated that secondary mineral formation is possible, but not direct quantification of precipitates has been performed."*

Line 783: I would restructure the sentence, as temperatures between 0 and 10 °C are actually quite common for coastal systems.

*We agree with the reviewer. That said, the range reported here was to represent coastal systems around Earth. Indicating that even in warm coastal waters serpentinization would still be unlikely. We have adapted the text on line 633-634:*

*However, serpentinization is unlikely during mERW as the process likely is negligeable at the temperature range of coastal systems (0–35°C)*

Lines 789-790: I suggest to delete the 'for soils' at the end of the sentence to avoid duplication of the word soil.

*Line 641: Agreed, we have removed the first mention of "soils", now reading:*

*"Even though the formation of saponite and iddingsite during olivine dissolution has been historically well-documented for soils"*

Line 945 and 950-952: see my third comment about secondary mineral formation.

Line 752-754 (former line 945): Changed sentence to:

*Most mERW experiments have been conducted in oxygenated seawater and are not fully representative of sedimentary conditions (e.g. in terms of $O_2$ and pH), and studies in sediments under hypoxic-anoxic bottom water have not directly measured the formation of secondary minerals (Fuhr et al., 2024).*

Line 758-759 (former 950-952): We have clarified that these authors have not quantified secondary mineral formation.

**Reviewer 2:**

Note, the line references by the reviewer are for the accepted changes document, we reported changes in reference to the new track-change document.

Line 13: No "peer-reviewed" results are available

Line 13: Suggestion implemented.

L185: For references on ecological impact, I would recommend adding more recent papers that provide empirical results such as Guo et al., 2024, Jankowska et al., 2024, and Hutchins et al., 2023. If this is too many references, consider cutting Bach et al., 2019 as this paper was just conceptual (i.e. didn't provide actual data)

Line 224: we have changed the references accordingly.

L187: The Vesta project has concluded (not in the first stage of execution).

Line 187: Correct, we have changed the sentence to: *"..., or are in the first stage of execution or results are being analyzed (e.g., Cornwall, 2023; USGS, 2023; Vesta, 2023)"*

L222: Would reword this given that the Cr is not contained within olivine, but rather within chromite. So, dunite contains Cr.

Line 223: We have changed "olivine" to "olivine-rich rock (dunite)".

L227: Again, recommend better references here. See above comment.

Line 227: changed accordingly.

L324-L325: This is overly simplistic. The likelihood of saturation is a balance between porewater exchange rate AND dissolution rate. Meaning, for the same porewater exchange rate, smaller particles, which dissolve faster, will drive saturation faster than larger particles.

Lines 327-328: The reviewer is correct that application parameters, such as grain size and the field specific dissolution rate will determine the rate at which dissolution products may build up. However, the aim of lines 327-328 was not to discuss when saturation effects occur, but rather to highlight that saturation are assumed not to occur in all three application scenarios. We later, explain that this assumption likely will not hold for cohesive sediments where the porewater exchange rate is lower than that of other sediment types. Nevertheless, we have clarified this further on line 328, now stating:

*"However, saturation effects are expected to occur in cohesive sediments with little advection or biological irrigation, or when dissolution rates are very high (e.g. when small grain sizes are used)."*

L361: You note incongruent mineral dissolution as one reason for spread in data. It could be good to include a sentence on the potential research avenue of improved olivine dissolution rate quantification using Si isotopes, rather than Si concentrations. The Si isotope method is a well established approach to measure silicate mineral dissolution rates while removing the confounding factor of secondary clay formation. To the best of my knowledge, this has not been done for olivine before.

Line 365: We would like to thank the reviewer for the suggestion. We have amended this to line 365-369, which now reads:

*"The problem of incongruent dissolution as well as the formation of secondary clay minerals could be potentially constrained through silicon isotope analysis (Chemtob et al., 2015; Gruber et al., 2013), if the source dunite rock has an isotope that is sufficiently distinct from the silicate sources in the application site sediment. However, such isotope analysis has not yet been performed in mERW studies and could be an avenue for future research."*

L404-405: Looking at Figure 4, it seems there are plenty of data on olivine dissolution at porewater pHs.

Indeed, there are dissolution rates at pore water pH values, but not in saltwater conditions, which are relevant for mERW. We have clarified on line 412-413:

*"The lack of data on olivine dissolution rates at pore-water pH in marine conditions and the absence of suitable spatial maps…"*

L525: Bearat et al. 2006 empirically tested particle abrasion of olivine as well and found it make a big difference.

Lines 531-534: The reviewer is correct that Béarat et al. 2006, also discusses the effect of grain abrasion, however, we opted not to include this reference here for two reasons. Firstly, this study focused on carbonation of minerals, which is a method where (in this case) olivine is exposed to high temperatures and pressure. Secondly, passivating mineral formation is much more pervasive in carbonation experiments than they are in mERW studies. We have amended lines 539-541, to include the experiments of Béarat et al., (2006), reading: *"Particle abrasion of olivine has also been found to strongly influence the rate of mineral carbonation at high temperature and*

*pressure, where the formation of passivating layers is likely more pervasive compared to mERW (Béarat et al., 2006)."*

Table 3: My reference equations for these reactions give different values for YAT. Also relevant for section 3.2.1 (e.g. L590).

Serpentine + Chrysotile + Lizardite: YAT = 0.25

$2\ Mg_2SiO_4 + H_2O + 2\ H^+ \rightarrow Mg_2^+ + Mg_3Si_2O_5(OH)_4$

Talc: YAT = 0.3

$4\ Mg_2SiO_4 + 10\ H^+ \rightarrow 5\ Mg_2^+ + 4\ H_2O + Mg_3Si_4O_{10}(OH)_2$

*We derived the serpentinization equations from Griffioen, (2017). A critical difference to the equation provided by the reviewer and the one we used by Griffioen is that in the reaction of Griffioen (2017) brucite is formed. The original reaction reported by Griffioen (2017) reads:*

$2\ Mg_2SiO_4 + 3H_2O \rightarrow Mg_3Si_2O_5(OH)_4 + Mg(OH)_2$

*Assuming brucite immediately dissolves or is not formed we get a formula akin to the one provided by reviewer 2 (written in function of $H_2O$ and $OH^-$):*

$2\ Mg_2SiO_4 + 3H_2O \rightarrow Mg_3Si_2O_5(OH)_4 + Mg^{2+} + 2OH^-$

*In the latter case, alkalinity is produced as represented by the 2 moles $OH^-$ produced. this means $\gamma_{AT}$ equals to 0.25, however when brucite is formed instead, $\gamma_{AT}$ equals 0. We have clarified the potential role of brucite now more clearly in the text, line 755 now reads: "When the brucite formed during serpentinization dissolves further, $\gamma_{AT}$ increases accordingly to 0.25".*

*In the case of talc formation, no reactions were reported as talc formation from olivine proceeds via or co-occurs with serpentine formation (as discussed on lines 788-791). Nevertheless, we have added talc formation as a reaction to Table 2. The value of $\gamma_{AT}$ we derive is different from the reviewer as talc formation produces 10 mol alkalinity following 4 moles olivine. Ideally 4 mol olivine would produce 16 mol alkalinity so that talc formation reduced the alkalinity production by a factor $\gamma_{AT}=0.625$ rather than 0.3.*

Sepiolite: YAT = 0.65

$6\ Mg_2SiO_4 + 16\ H^+ \rightarrow 8\ Mg_2^+ + H_2O + Mg_4Si_6O_{15}(OH)_2 \cdot 6H_2O$

Saponite (x ≈ 0.3-0.6, using x = 0.5 illustratively): YAT = 0.43

$3.5\ Mg_2SiO_4 + 0.25\ Ca_2^+ + 0.5\ Al_3^+ + 6H^+ \rightarrow 4\ Mg_2^+ + (2\text{-}n)\ H_2O + Ca_{0.25}Mg_3Al_{0.5}Si_{3.5}O_{10}(OH)_2 \cdot n\ H_2O$

*The equations used for sepiolite and saponite were taken from Isson and Planavsky, (2018). We are grateful to the reviewer in taking note that $\gamma_{AT}$ in both reactions was incorrectly referenced; we had overlooked the need to subtract the alkalinity consumption caused by precipitation from*

*the alkalinity produced during the primary weathering reactions when the tables were merged. As such the correct $\gamma_{AT}$ are 0.67 and 0.43 for sepiolite and saponite formation respectively, rather than 0.33 and 0.60. Following the comments of reviewer 2, we have carefully revised the table formulas and references again. To this end, we noted that a reference to Griffioen (2017) for serpentine formation was inadvertently removed from the first version when merging the tables. The reference is now restored again.*

L751: I would add a reference for the new Zhou et al., 2024 paper, and include its findings in your discussion.

*Line 769 and further: The reference to Zhou were included throughout the text. Since their results were very similar to the results of He and Tyka (2023), we have not discussed their results separately. However, we have added a critical note on their findings and how they translate to mERW specifically line 775-779 now reads:*

*"High values for $\eta_{AT}$ were also associated with highly stratified systems, since the surface-released alkalinity remains in contact with the atmosphere for longer (He and Tyka, 2023; Zhou et al., 2024). However, these model results cannot be directly applied to mERW, since alkalinity is released from the sediment, and not directly within the surface water, and so high stratification would prevent sediment-borne alkalinity of reaching the surface."*